

# Measurement Report: Understanding the seasonal cycle of Southern Ocean aerosols

Ruhi S. Humphries[1,2], Melita D. Keywood[1,2], Jason P. Ward[1], James Harnwell[1], Simon P. Alexander[3,2], Andrew R. Klekociuk[3,2], Keiichiro Hara[4], Ian M. McRobert[5], Alain Protat[6,2], Joel Alroe[7], Luke T. Cravigan[7], Branka Miljevic[7], Zoran D. Ristovski[7], Robyn Schofield[8], Stephen R. Wilson[9], Connor J. Flynn[10], Gourihar R. Kulkarni[11], Gerald G. Mace[12], Greg M. McFarquhar[13], Scott D. Chambers[14], Alastair G. Williams[14], and Alan D. Griffiths[14]

[1]Climate Science Centre, CSIRO Oceans and Atmosphere, Melbourne, Australia
[2]Australian Antarctic Program Partnership, Institute for Marine and Antarctic Studies, University of Tasmania, Hobart, Tasmania, Australia
[3]Australian Antarctic Division, Channel Highway, Kingston, Tasmania, Australia
[4]Department of Earth Science System, Faculty of Science, Fukuoka University, Jyonan, Fukuoka, Japan
[5]Engineering and Technology Program, CSIRO National Collections and Marine Infrastructure, Hobart, Australia
[6]Australian Bureau of Meteorology, Melbourne, Australia
[7]School of Earth and Atmospheric Sciences, Queensland University of Technology, Brisbane, Australia
[8]School of Geography, Earth and Atmospheric Sciences, University of Melbourne, Parkville, Victoria, Australia
[9]Centre for Atmospheric Chemistry, School of Earth, Atmospheric and Life Sciences, University of Wollongong, Wollongong, New South Wales, Australia
[10]School of Meteorology, University of Oklahoma, Norman, United States of America
[11]Atmospheric Sciences and Global Change Division, Pacific Northwest National Laboratory, Richland, United States of America
[12]Department of Atmospheric Science, University of Utah, Salt Lake City, United States of America
[13]Cooperative Institute for Mesoscale Meteorological Studies, University of Oklahoma, Norman, United States of America
[14]Environmental Research, ANSTO, Lucas Heights, New South Wales, Australia

**Correspondence:** Ruhi Humphries (Ruhi.Humphries@csiro.au)

**Abstract.** The remoteness and extreme conditions of the Southern Ocean and Antarctic region have meant that observations in this region are rare, and typically restricted to summertime during research or resupply voyages. Observations of aerosols outside of the summer season are typically limited to long-term stations, such as Kennaook/Cape Grim (KCG, 40.7°S, 144.7°E) which is situated in the northern latitudes of the Southern Ocean, and Antarctic research stations, such as the Japanese operated

5    Syowa (SYO, 69.0°S, 39.6°E). Measurements in the mid-latitudes of the Southern Ocean are important, particularly in light of recent observations that highlighted the latitudinal gradient that exists across the region in summertime. Here we present two years (March 2016 - March 2018) of observations from Macquarie Island (MQI, 54.5°S, 159.0°E) of aerosol (condensation nuclei larger than 10 nm, $CN_{10}$) and cloud condensation nuclei (CCN at various supersaturations) concentrations. This important multi-year data set is characterised, and its features are compared with the long-term data sets from KCG and SYO

10    together with those from recent, regionally relevant voyages. $CN_{10}$ concentrations were the highest at KCG by a factor of ~50% across all non-winter seasons compared to the other two stations which were similar (summer medians of 530 $cm^{-3}$, 426 $cm^{-3}$ and 468 $cm^{-3}$ at KCG, MQI and SYO, respectively). In wintertime, seasonal minima at KCG and MQI were sim-



ilar (142 $cm^{-3}$ and 152 $cm^{-3}$, respectively), with SYO being distinctly lower (87 $cm^{-3}$), likely the result of the reduction in sea spray aerosol generation due to the sea-ice ocean cover around the site. $CN_{10}$ seasonal maxima were observed at the stations at different times of year, with KCG and MQI exhibiting January maxima and SYO having a distinct February high. Comparison of $CCN_{0.5}$ data between KCG and MQI showed similar overall trends with summertime maxima and wintertime minima, however KCG exhibited slightly ($\sim$10%) higher concentrations in summer (medians of 158 $cm^{-3}$ and 145 $cm^{-3}$, respectively), whereas KCG showed $\sim$40% lower concentrations than MQI in winter (medians of 57 $cm^{-3}$ and 92 $cm^{-3}$, respectively). Spatial and temporal trends in the data were analysed further by contrasting data to coincident observations that occurred aboard several voyages of the RSV Aurora Australis and the RV Investigator. Results from this study are important for validating and improving our models, highlight the heterogeneity of this pristine region, and the need for further long-term observations that capture the seasonal cycles.

## 1 Introduction

Understanding the pre-industrial atmosphere is fundamental to characterising the impact of anthropogenic activity on the world's climate. Unlike other atmospheric species (e.g. greenhouse gases), atmospheric time capsules such as ice cores do not exist for many aerosol properties. To gain an understanding of this baseline atmosphere, we need to measure a proxy atmosphere that is as near to being anthropogenic-free as is possible. The Southern Ocean and Antarctic region is one of the most remote regions, being far from both continental and urban influences. This makes it an important location to understand the pre-industrial atmosphere and the natural processes that occur here that are often masked by the much larger signals associated with anthropogenic activity (Carslaw et al., 2013; McCoy et al., 2020).

The most recent Intergovernmental Panel on Climate Change (IPCC) reports (IPCC, 2014, 2021) identified aerosol-cloud interactions as exhibiting the largest uncertainties in our understanding of the Earth's climate. A large driver for this is the natural background processes that can only be observed in pristine regions, as illustrated by the significant uncertainties and biases that exist in the simulation of clouds, aerosols and air-sea exchanges in climate and earth system models in the Southern Ocean region (e.g. Marchand et al., 2014; Shindell et al., 2013; Pierce and Adams, 2009). Importantly, these biases are driven by a poor understanding of the underlying physical processes occurring in the region, with impacts seen in our understanding of tropical rainfall distribution (Frey and Kay, 2018), the global energy budget (Trenberth and Fasullo, 2010), and our ability to model the impact of carbon-cycle and cloud feedbacks on climate change (IPCC, 2014; Gettelman et al., 2016).

Aerosols in this region are well known to be derived primarily from natural sources, including primary particles (sea spray and bubble bursting), which dominate the mass concentration, and secondary particles, which drive the number concentrations of both condensation nuclei (CN) and cloud condensation nuclei (CCN). Numerous observations (e.g. Shaw, 1988; Gras, 1983; Bigg et al., 1984; Kreidenweis et al., 1998; Bates et al., 1998; Covert et al., 1998; Quinn et al., 2000; Rinaldi et al., 2010, 2020; Frossard et al., 2014; Sanchez et al., 2018) have found that the secondary particles in the region originate from non-sea salt sulfate, likely originating from the oxidation products of dimethyl sulfide (DMS) emitted from phytoplankton and sea-ice algae, such as methanesulfonic acid (MSA) and sulfuric acid. Seasonal emissions of these aerosol precursor gases coincides with the





significant seasonal cycles of phytoplankton populations (Lana et al., 2011), resulting in the typical annual cycles observed in aerosol populations in the region.

Phytoplankton abundance exhibits strong regional heterogeneity and latitudinal gradients, with the highest concentrations centered near the sea ice regions (Deppeler and Davidson, 2017). This spatial heterogeneity, combined with the atmospheric
transport processes and timelines for chemical and physical generation and processing of aerosols, is likely to result in significant spatial heterogeneity in aerosol populations across the region that are of a similar magnitude, but may not be well correlated with phytoplankton abundance.

Intensive field campaigns with aerosol observations have occurred only a handful of times throughout the last century (Bigg, 1990a; Bates et al., 1998; O'Dowd et al., 1997; Boers, 1995). However, in response to the recognition of the importance of
the region to the climate and Earth system more widely there has been a recent flurry of aerosol observations in the region, resulting in at least 16 aerosol inclusive campaigns between 2009 and 2018 (Wofsy, 2011; Law et al., 2017; Humphries et al., 2015, 2016; Dall'Osto et al., 2017; Fossum et al., 2018; Stephens et al., 2018; Schmale et al., 2019; Brock et al., 2019; Sato et al., 2018; McFarquhar et al., 2021; Mace and Protat, 2018; Mace et al., 2021; Alroe et al., 2020a; Simmons et al., 2021; Sanchez et al., 2021; Twohy et al., 2021), with at least another 20 more campaigns planned in the region before the end of
2025. Of particular note is the establishment of ongoing observations (as part of the World Meteorological Organisation's Global Atmosphere Watch, WMO-GAW, programme) aboard Research Vessel Investigator, based out of Australia, and the soon to be online observations aboard the French owned Marion Dufresne II.

While ongoing observations are particularly important for characterising seasonal and long-term changes in the region, the challenges associated with ongoing operation in this environment make them few and far between. There have been many
observational programs incorporating aerosol microphysical measurements across the region that have spanned from a few weeks, to many years (e.g. Asmi et al., 2010; Koponen et al., 2003; Kyrö et al., 2013; Claeys et al., 2010; Kubicki et al., 2016; Savoie and Prospero, 1989; Li et al., 2018; Schmale et al., 2013; Bigg et al., 1983; Gras, 1993; Pant et al., 2011; Hansen et al., 2009; Brechtel et al., 1998; Weller et al., 2018). Observational platforms that are currently in operation and present long-term records include those in the East Antarctic sector of the continent or Southern Ocean: Syowa (69.0°S, 39.0°W; e.g. Hara
et al., 2011b) and Kennaook/Cape Grim (40.7°S, 144.7°E; e.g. Gras and Keywood, 2017); those in the West Antarctic sector: Neumayer (70.67°S, 8.27°W; e.g. Weller et al., 2015, 2011), King Sejong Station (62.22°S, 58.78°W; e.g. Hong et al., 2020; Jang et al., 2022) and Princess Elisabeth Station (71.95°S, 23.35°E; e.g. Herenz et al., 2019); and South Pole (0°E, 90°S; e.g. Park et al., 2004). While this list might sound long, many of these observations are patchy over the many decades since this technology has been viable, and with only five stations undertaking ongoing continuous observations over such a large and
heterogeneous spatial area, it is striking how little data exists in a region that is so important for both the regional and global climate.

In this study, we focus primarily on observations in the East Antarctic region, presenting newly acquired data from Macquarie Island, and contrasting these observations with those from nearby East Antarctic observations at long-term stations (i.e. Kennaook/Cape Grim, and Syowa), as well as numerous recent ship voyages in the region. Observations of aerosol properties
at Macquarie Island have occurred previously during the ACE 1 campaign for approximately three weeks in late spring/early

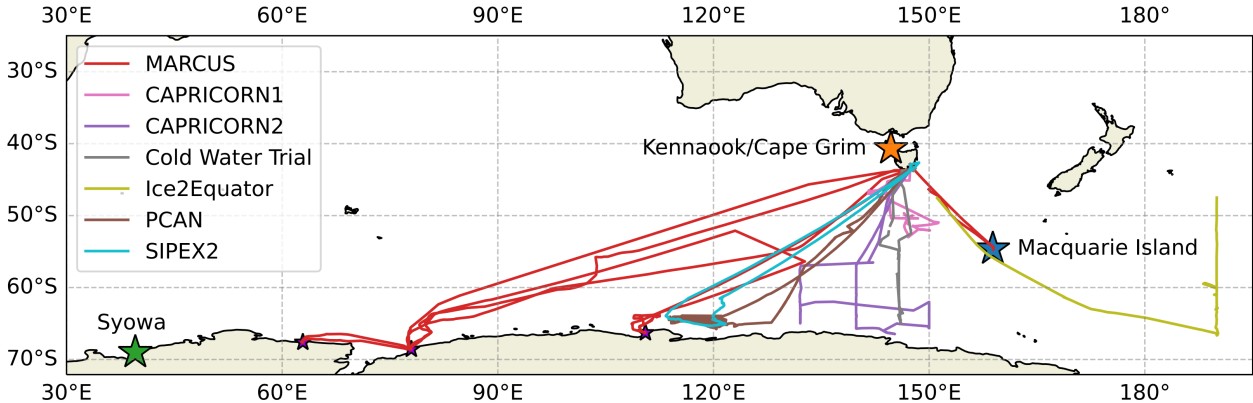

**Figure 1.** Map showing the relative locations of the three long term stations utilised in this study: Kennaook/Cape Grim (40.7°S,144.7°E), Macquarie Island (54.5°S, 159.0°E) and Syowa (69.0°S, 39.6°E). Voyage tracks of those voyages utilised in this study are also shown. Both SIPEX2 and MARCUS occurred aboard ice breaker RSV Aurora Australis, with the latter occurring throughout a full season of four resupply voyages to Australian Antarctic stations (from west to east (not labelled): Mawson, Davis, Casey and Macquarie Island). Note that data from the Ice2Equator voyage have been subselected (> 47.5°S) for those likely to be representative of Southern Ocean airmasses unaffected by New Zealand continental influence or tropical air further north.

summer of 1995 (Brechtel et al., 1998; Kreidenweis et al., 1998). These represent valuable observations and had important results including that aerosol properties depended largely on the airmass origin, with influences from both Antarctica and Tasmania, as well as clean marine air originating over the Southern Ocean. This study builds on that short campaign, and extends previous work undertaken characterising the seasonal changes in Southern Ocean aerosol populations (e.g. Gras and Keywood,

2017; Weller et al., 2011; Bigg et al., 1983; Gras, 1993, 1990; Hara et al., 2011b; Kim et al., 2017), as well as further expanding on work undertaken by Alroe et al. (2020b) and Humphries et al. (2021a) in understanding the latitudinal changes observed across this region. In particular, this work takes both seasonal station data and voyage data to develop an understanding of both seasonal and latitudinal variability across the East Antarctic sector of the Southern Ocean region.

## 2 Methods

Two years of observations of $CN_{10}$ and CCN were made at the Australian sub-Antarctic research station located on Macquarie Island in the mid-latitudes of the Southern Ocean (54.5°S, 159.0°E). To support the analysis of these data and to give greater spatial and temporal context to the data set, observations from a number of other long-term stations and campaigns are utilised in this study. Figure 1 shows the locations of these observations. Aerosol data from three ground-based stations are included: Macquarie Island, Kennaook/Cape Grim (40.7°S,144.7°E) and Syowa (69.0°S, 39.0°E). In addition, data from a range of

intensive observational campaigns aboard two research vessels, the RSV Aurora Australis, and the RV Investigator have been utilised, and are detailed below.



## 2.1 Macquarie Island

Macquarie Island (MQI) is a small, isolated island (34 km long and up to 5 km) oriented approximately north-south in the Southern Ocean. A comprehensive site description is provided separately by Stavert et al. (2019). Aerosol observations at MQI were made in the Clean Air Laboratory, located on an isthmus near the northern tip of the island, and upwind (westward) of the adjacent permanently-occupied research station. The observational program ran as part of the MICRE (Macquarie Island Cloud and Radiation Experiment) and ACRE (Antarctic Clouds and Radiation Experiment) projects from March 2016 to March 2018 (McFarquhar et al., 2021; Tansey et al., 2022). Together these projects deployed a suite of instrumentation including cloud lidars and radars, a distrometer, a range of radiometers, and most relevant to the current study, in-situ aerosol samplers measuring condensation nuclei (CN) and cloud condensation nuclei (CCN) number concentrations. All CN and CCN data from this project are available at Humphries et al. (2021c).

### 2.1.1 Condensation Nuclei

Number concentrations of condensation nuclei (aerosols) larger than 10 nm ($CN_{10}$) were measured continuously at 1 Hz using a condensation particle counter (CPC Model 3772, TSI Inc. Shoreview, MN, USA). As is the case with all TSI CPCs, the CPC draws sample air continuously at a pre-defined flow rate through a chamber where the air is supersaturated with 1-butanol, providing an environment where the vapour will condense onto aerosols as small as 10 nm (with default manufacturer's settings for this model), growing them to supermicron sizes where they are counted individually using a simple optical particle counter. The instrument comes preinstalled from the factory with an internal critical orifice to maintain the nominal flow rate at $1.0\ \mathrm{L.min^{-1}}$. However in marine environments, this critical orifice can block with sea salt and requires regular cleaning to prevent flow rate drift outside acceptable limits. Given the remote operation of these instruments, the CPC was reconfigured by replacing the critical orifice with an active mass flow controller (MFC; Alicat Scientific Model MC 5LPM) set to $1.0\ \mathrm{L.min^{-1}}$. The MFC was calibrated in-situ using an external low-pressure drop flowmeter (Sensidyne Gilibrator, St. Petersburg, FL, USA) placed in front of the CPC. MFC data were captured via the analog input of the CPC for logging. Data from the CPC were filtered for periods of instrument zeros, flow checks and other maintenance and outages (major outages detailed below). Once a clean data set was obtained, hourly statistics were calculated for the full multiyear data set.

There were two significant periods where data were removed. The first occurred from the start of measurements in March 2016 up until 16 June 2016 where the nozzle pressure increased slowly out of instrument specification due to sea salt aerosol build-up in the nozzle. While nozzle cleaning was already part of the scheduled maintenance, the build-up occurred much faster than anticipated. This resulted in a noticeable artifact in the CCN/CN ratio that ended up being well above 1 (at its worst). All data were removed during this period and it was verified that this issue did not reoccur during the remainder of the campaign. The second issue occurred due to issues with the synchronisation with the time server. While much of these data were able to be recovered appropriately, data from 2 June to 21 July 2017 were too far out of sync with real time, and were removed from the data set.



### 2.1.2 Cloud Condensation Nuclei

Number concentrations of CCN were measured continuously at 1 Hz at a range of supersaturations for the entire two years of observations using a continuous-flow, streamwise thermal-gradient CCN counter (CCNC; Model CCN-100, Droplet Measurement Technologies, Longmont, CO, USA). Because of logistical constraints preventing scientific experts visiting the island for annual calibrations, the instrument was swapped out by onsite technicians after one year for a model that had been recently serviced and calibrated by the manufacturer. While instrument intercomparison could not be undertaken for this project,

ongoing intercomparison of different instruments of the same model at Kennaook/Cape Grim have shown good agreement between instruments. The instrument was initially configured to run at 0.5% supersaturation for 23 hours of the day, then in the remaining hour, move sequentially through a full set of six supersaturations (10 minutes at each), including (in order): 1.0%, 0.8%, 0.6%, 0.5%, 0.4%, and 0.2%. On 2 November 2016, this sampling pattern was altered so that it would scan through the supersaturation sequence every hour of the day.

Calibration of the supersaturation settings of both instruments deployed during ACRE was undertaken at the Droplet Measurement Technologies workshop in Boulder, Colorado, which is at a significant altitude above sea level (1620 m above sea level, 830 mbar). A supersaturation pressure correction was applied, resulting in actual measured supersaturations being 0.055% higher than set points (e.g. 0.2% was actually 0.255%). Although a range of supersaturations were measured during this campaign (stated above), because of the need to compare observations with other sites where often only 0.5%

supersaturation was measured, this study focuses largely on this supersaturation. It is important to note though that a number of recent studies (Fossum et al., 2018; Sanchez et al., 2021; Twohy et al., 2021) have suggested that 0.3% is likely to be the most representative of actual environmental conditions in this region.

Data were thoroughly quality controlled. The first three minutes of each 10 minute period were removed to allow for the stabilisation of instrument conditions. Remaining data were screened for periods when instrument parameters were out of man-

ufacturer specification, instrument maintenance and other outages. After this quality control, hourly statistics were calculated for each supersaturation. One major instrument malfunction occurred during the campaign: the internal nafion membrane that provides the essential supersaturation conditions within the instrument rapidly degraded over a period of about 7 weeks. This resulted in removal of data from 25 April 2017 until 17 July 2017, at which point the nafion was replaced.

Another major quality control issue arose because the CPU clock housed within the embedded PC of the CCNC exhibited

significant time drift. Although time syncing with server time was performed weekly (and increased to daily after identification of this issue), the magnitude of the time drift was such that it still presented a major issue. Since the CCN population is a subset of the CN population, many features were observable in both data sets, allowing quantification of the time drift through comparison of the two time series. Unfortunately, this comparison revealed that the time offset drifted both forward and backward in time, and could not be corrected for. Data were removed entirely when this drift became significant (greater

than 2 minutes per day), as occurred between the August 19 and 25, 2017. With the remaining data set, there were seven distinct periods where the time drift was different and highly variable, and this varied from 4.5 minutes slow, to 3.25 minutes fast. Because the goal was to produce hourly statistics for these data, it was decided (after numerous failed attempts to align



the data with various mathematical methods) that data at the start and end of each hour would be removed prior to resampling to hourly statistics. This ensured that the time drifting issue had no affect on resampled data.

### 2.1.3 Sampling system

Sample air was drawn into the Clean Air Laboratory through a sample inlet positioned 1.5 m above the roofline ($\sim$4 m above ground level, 10 m above sea level). The sample line was a bespoke design with a TSP (Total Suspended Particles) weather hat mounted atop a vertical 3" OD stainless tubing which provided both protection and vertical stability in the high wind environment. Through a bored-out Swagelok bulkhead fitting, a ¾" OD stainless steel tube penetrated through the external 3" OD tube to the height of the weather hat. The weather hat had three tie-down holes for guy wires to be attached for stabilisation to the laboratory roof. The top metre of the inlet, together with the weather hat were heated continuously to prevent icing with self-regulating heater tape designed to prevent ice build-up on commercial walk-in freezer doors. Heater tape was covered in insulation, following by weather and UV-proof heat-shrink.

The ¾" tubing that penetrated through the roof was connected inside the lab to a PM2.5 cyclone (URG Corp, Chapel Hill, NC, USA), with a sample flow rate of 16.67 L.min$^{-1}$. Both the CCNC and CPC sampled air at flow rates of 0.5 L.min$^{-1}$ and 1.0 L.min$^{-1}$, respectively. In addition, there was a bypass flow of 15.1 L.min$^{-1}$ set by a variable area flow meter. Before the CCNC, a buffer volume was attached off the side of a T-piece to help dampen pressure fluctuations caused by outside wind moving across the inlet which the instrument can be sensitive to. To prevent water contamination of the 1-butanol working fluid, the sample was dried prior to entering the CPC using a nafion drier (Nafion Drier model MD-110-12S-4, Perma Pure LLC, Lakewood, NJ, USA) setup using the differential pressure configuration with 1 L.min$^{-1}$ sample flow (defined by the CPC's MFC), and 5 L.min$^{-1}$ of HEPA filtered laboratory air as sheath flow (defined by a rotameter).

## 2.2 Kennaook/Cape Grim

The Kennaook/Cape Grim (KCG) baseline atmospheric programme is one of three premier global stations of the WMO-GAW programme. The observatory is located at the northwest tip of Tasmania (40°41'S, 144°41'E) atop a cliff, sampling 94 m above sea level. The location enables the observation of Southern Ocean air that has had minimal recent anthropogenic influence. Air sampled in the "baseline" sector (190-280°) has typically spent the previous several thousand kilometres without contact with land. Previously known as Cape Grim, the Indigenous heritage of the area has recently been recognised by the adoption of a dual-naming convention. With enthusiastic support from the community and station staff and scientists, the station is now known as Kennaook/Cape Grim.

Aerosol measurements at KCG have been occurring since the mid-1970s with a range of technologies, and has followed what have now been established as WMO-GAW Aerosol Programme recommendations. In this study, data between 2010 and 2020 (inclusive) have been utilised for both CN$_{10}$ (CPC Model 3010, TSI Inc. Shoreview, MN, USA) and CCN$_{0.5}$ (Model CCN-100, Droplet Measurement Technologies, Longmont, CO, USA). The most recent analysis of these long-term data sets, along with a further details of the measurements, is provided by Gras and Keywood (2017). Hourly median concentration data (CCN and CN$_{10}$) are available in the World Data Centre for Aerosols (http://www.gaw-wdca.org/). Note that only data from



the baseline sector are used in this study to ensure air from the continent and anthropogenic regions is kept to a minimum. The baseline sector here is defined as periods with wind directions between 190° and 280°, as well as radon concentrations below 100 mBq.

## 2.3 Syowa

Syowa Station (SYO) is located on East Ongul Island in Lutzow Holm Bay in East Antarctica (69.0°S, 39.0°E). The station has a significant sea ice presence around the station, being approximately 100 km in summer, and 1000 km in the winter-spring period. Comprehensive details of the station and the aerosol program are provided in previous publications (e.g. Hara et al., 2011b, a).

Observations of aerosol number concentrations have been made continuously at SYO since 1997 using multiple CPCs (long-
term record has utilised CPC Model 3010, TSI Inc. Shoreview, MN, USA). In 2004, a new clean air observatory was built at the station to replace the aging observatory. Simultaneous measurements were undertaken in both observatories (using the same TSI Model 3010 CPC) during the 2004 period to ensure continuity of the record. While agreement is good between the records, the aerosol inlet in the new clean air observatory is better designed. For the purposes of the current study, only ∼10 years of data were desired to enable a climatological comparison with the MQI data. Consequently, only data available from the new
clean air laboratory were utilised in this study, which included data from 2004 to 2016. All data are publicly available at Hara et al. (2022).

## 2.4 Voyages aboard the RSV Aurora Australis

The RSV Aurora Australis (retired in 2020) was Australia's flagship ice-breaker designed to undertake both marine science and station resupply voyages to the Antarctic continent. The vessel had no dedicated atmospheric composition observational
capability, and consequently, instrumentation was installed for specific campaigns, including full inlet systems.

### 2.4.1 MARCUS

The MARCUS (Measurements of Aerosols, Radiation and CloUds over the Southern Oceans) campaign occurred as a project aboard the RSV Aurora Australis during its summer resupply of Australia's Antarctic research stations between October 2017 and March 2018 (McFarquhar et al., 2021; Alexander and Klekociuk, 2021). Full details of the campaign have been published
separately (Humphries et al., 2021a). In short, the campaign utilised the United States Department of Energy (DOE) Atmospheric Radiation Measurement (ARM) Program Mobile Facility 2 (AMF2). Cloud, radiation and precipitation instruments were deployed, along with the Aerosol Observing System (AOS) (https://www.arm.gov/capabilities/instruments/aos). Utilised in this study are the $CN_{10}$ (CPC Model 3772, TSI Inc. Shoreview, MN, USA) and CCN observations (Model CCN-100, Droplet Measurement Technologies, Longmont, CO, USA). Note that because of the lack of dedicated inlet systems for air sampling,
campaign specific systems were installed and due to operational limitations, were located directly adjacent to the ship's own exhaust. This meant that this data set was heavily contaminated by the platform's own diesel exhaust, and consequently just





over 10% from the campaign was deemed usable (Humphries et al., 2021a). Exhaust-free aerosol data from the MARCUS campaign are available at Humphries (2020), with raw data available at the ARM archive (references within).

### 2.4.2 SIPEXII

The second Sea Ice Physics and Ecosystems eXperiment (SIPEXII) occurred from 14 September to 11 November in 2012. The 52 day marine science voyage travelled from its home port in Hobart, to the East Antarctic pack ice between 112°E and 122°E during which time it setup eight temporary research stations atop ice-floes (1-5 day periods anchored to the drifting ice-floe). Full details of atmospheric observations during SIPEXII are found in earlier publications (Humphries et al., 2015, 2016). Aerosol measurements were limited to in-situ aerosol number concentrations using two concentration particle counters (CPCs)
with different size cuts of particles larger than 3 nm ($CN_3$; Model 3025A, TSI, Shoreview, MN, USA) and 10 nm ($CN_{10}$; Model 3772, TSI, Shoreview, MN, USA). SIPEXII data are incorporated in this study to compare with the MQI observations, and consequently, only $CN_{10}$ data are utilised here. Aerosol data from SIPEXII can be found at Humphries et al. (2014). Aerosol inlets for this voyage were able to be located further from the platform's exhaust compared with MARCUS, resulting in a greater percentage of data that were deemed usable due to contamination from the exhaust.

### 2.5 Voyages aboard the RV Investigator

The RV Investigator is Australia's flagship blue-water research vessel. Its home port is in Hobart and since its commissioning in 2015, has spent about two-thirds of its research voyage time in the Southern Ocean. It has two dedicated atmospheric laboratories, home to a suite of instrumentation for the measurement of aerosols, reactive gases and greenhouse gases. These instruments operate continuously throughout the year, and have resulted in the platform being recognised as the first mobile
platform of the WMO-GAW programme in 2018. Further details of the platform are provided in Humphries et al. (2019). Unfortunately, the data from all voyages of the RV Investigator is still being quality controlled, with all data from 2015 to 2021 (inclusive) expected to be published in the near future. Data from selected voyages, however, is already available, and those data sets utilised in this study are outlined below.

Unlike the RSV Aurora Australis, the RV Investigator's atmospheric laboratories are connected to a purpose build air sam-
pling inlet. This inlet is situated at the fore of the vessel, approximately 18 m above sea level, and is as far away from the ship's exhaust stack as is possible on the vessel. This scenario, combined with the different operational constraints of the vessel (i.e. greater freedom to align the vessel with its nose into the wind, rather than orienting to suit the ice conditions, as was often the case for the RSV Aurora Australis), means that contamination of atmospheric data by platform exhaust is much lower on the RV Investigator compared with the RSV Aurora Australis. Typically, exhaust contamination aboard the RV Investigator ranges
between 10 - 50% of the time.



### 2.5.1 Cold Water Trial

The Cold Water Trial voyage (voyage ID: IN2015_E01) occurred between January 29 and February 17 of 2015. This voyage was the first voyage of RV Investigator into polar waters, and occurred during its commissioning phase. The voyage travelled south along the 146th line of longitude, reaching the Antarctic ice edge at approximately 65°S. A comprehensive analysis of these data are provided by Alroe et al. (2020a). Both $CN_{10}$ (Model 3772, TSI, Shoreview, MN, USA) and $CCN_{0.5}$ (Model CCN-100, Droplet Measurement Technologies, Longmont, CO, USA) are available and were utilised for the current study. All data are available at Humphries et al. (2022b).

### 2.5.2 CAPRICORN1

The CAPRICORN (Clouds, Aerosols, Precipitation, Radiation, and atmospheric Composition Over the southeRn oceaN) campaign occurred aboard the RV Investigator between 14 March and 16 April 2016 (part of IN2016_V02, Mace and Protat (2018)). The voyage departed Hobart and spent its time undertaking a range of marine characterisation and sampling down to 53°S due south of Hobart. The campaign deployed a full suite of instrumentation, however included here is just CCN concentration measured at 0.5% supersaturation (Model CCN-100, Droplet Measurement Technologies, Longmont, CO, USA). Data are publicly available at Protat and Humphries (2020).

### 2.5.3 Ice2Equator

The Ice2Equator campaign occurred as part of the IN2016_V03 voyage, directly after the CAPRICORN2 voyage, between April 25 and June 30, 2016. The voyage travelled south-east from Hobart to the Antarctic ice edge (66.7°S), before heading north along the 170°W longitude line to the equator. The voyage concluded in Lautoka, Fiji, and also had a mid-voyage stopover in Wellington, New Zealand for personnel change-over. Full details of the voyage and results from its atmospheric analysis are provided in the PhD thesis of Alroe (2021), and are expected to be published in a peer-reviewed journal in the near future. Because the focus of the current study is just on background Southern Ocean air, data from this voyage have been limited to only those south of 47.5°S in order to both keep data representative of Southern Ocean airmasses, and ensure we avoid continental influence during periods downwind of New Zealand. $CCN_{0.5}$ concentrations (Model CCN-100, Droplet Measurement Technologies, Longmont, CO, USA) have been utilised in this study. All data are available at Humphries et al. (2022a).

### 2.5.4 PCAN

The Polar Cell Aerosol Nucleation (PCAN) campaign occurred as part of the IN2017_V01 voyage between January 14 and March 4, 2017. The voyage departed Hobart, traversing south-west across the Southern Ocean before spending most of its time (40 days) in the study area south of 60°S and between 110 and 120°E. Because one of the primary goals of the voyage was mapping the seafloor in the study area, a "mowing-the-lawn" pattern was adopted, which increased the impact of sampling platform exhaust, resulting in higher than usual exhaust contamination in these data. Despite this, data are still valuable and





CCN concentrations (Model CCN-100, Droplet Measurement Technologies, Longmont, CO, USA) at a range of supersaturations have been utilised in this study. Full details and analysis of the voyage data are provided in Simmons et al. (2021), and data are available at Humphries et al. (2020).

### 2.5.5 CAPRICORN2

The second CAPRICORN campaign occurred between January 11 and February 21 of 2018 (part of IN2018_V01) and ventured all the way to the Antarctic ice edge. Data from this voyage have been published previously in Humphries et al. (2021a). As in CAPRICORN1, a range of instrumentation was deployed, however we have limited our analysis here to those consistent with MQI observations - i.e. $CN_{10}$ (Model 3772, TSI, Shoreview, MN, USA) and CCN (Model CCN-100, Droplet Measurement Technologies, Longmont, CO, USA) at various supersaturations. These data are available at Humphries et al. (2021b).

## 3 Results and Discussion

The full time series of $CN_{10}$ and $CCN_{0.5}$ from MQI is show in Figure 2a. Consistent with previous observations in this region (e.g. Bigg, 1990b; Gras and Keywood, 2017), concentrations were observed to be very low (relative to other marine environments), exhibiting summer maxima and winter minima (shown in Figures 2b and 2c). The hourly resolution time series shows significant short-term spikes, many of which coincide with winds from the northerly direction (270° - 90°). Interestingly, the local research station is located directly north-east of the clean air observatory, and consequently, may only be able to explain a small percentage of the large excursions in the data. Given the majority of these spikes are not from local sources, it is reasonable to suggest that these events are likely the result of long-range transport from continental airmasses leaving mainland Australia. Correlation plots (not shown) of CN and CCN standard deviations with radon concentrations reveal that while some of this variability can be explained by continental influence, that most of the data, including the high aerosol concentrations, occur when radon concentrations are below 150 mBq, which is reflective of maritime air.

### 3.1 $CN_{10}$ seasonal cycles

Figure 2b shows the seasonal cycle of $CN_{10}$ from MQI alongside those from KCG and SYO. Note that the seasonal cycle is plotted over 18 months to show the maxima and minima clearly, is presented in Appendix Figure A1a. Summertime maxima are observed for all three stations, with the highest concentrations observed at the northernmost station, KCG. SYO exhibits summertime concentrations of a similar magnitude as KCG, however the timing of the maxima is offset by one month, peaking in February, as compared with the other two stations which peak in January. While the timing of the summertime maxima are the same for MQI and KCG, the concentrations observed at the latter are 10-40% higher. Concentrations at MQI and SYO remain high until a sudden drop in April, before their winter minima begins in May. In contrast, KCG concentrations begin declining immediately after the January peak, but make a more steady decline to reach the winter minima in June. Springtime increases begin almost simultaneously in September at both KCG and SYO, although the increase at MQI is comparatively





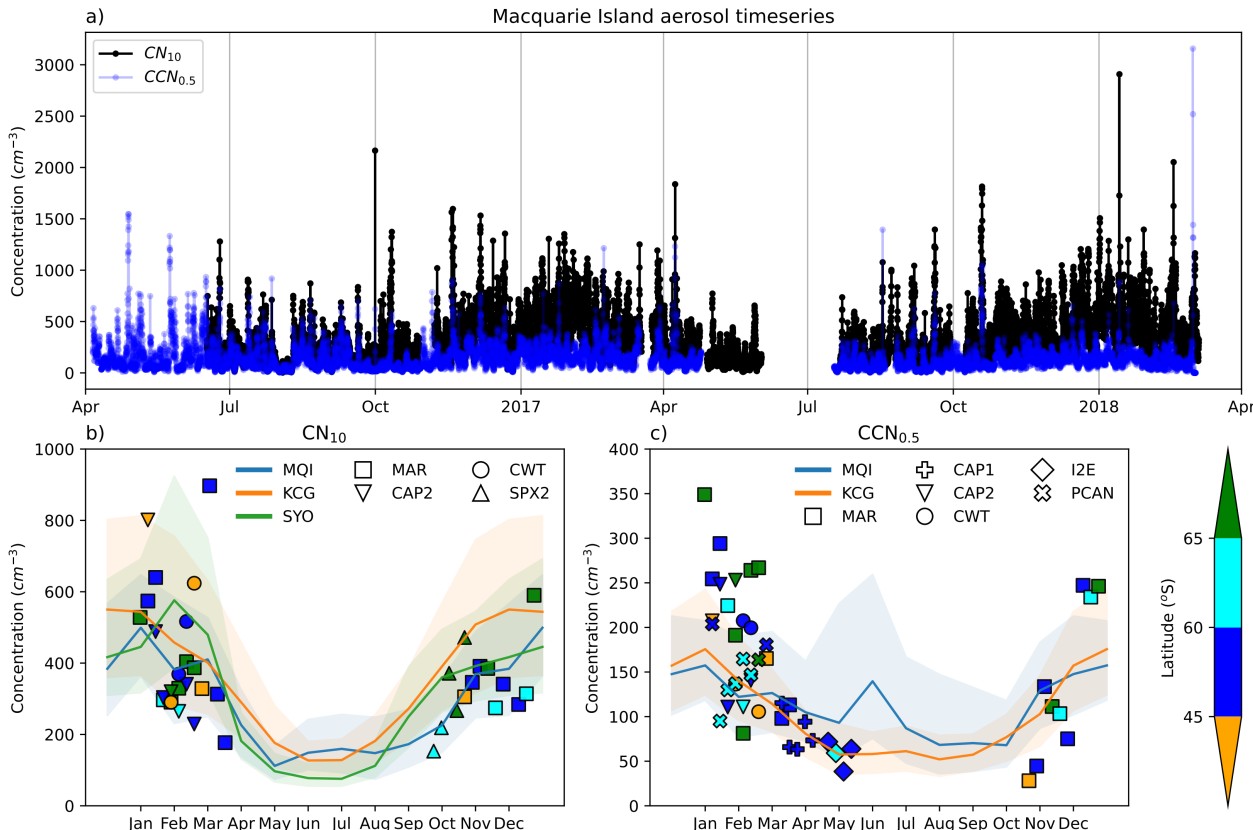

**Figure 2.** The full time series (a) of hourly median aerosol observations (only showing $CN_{10}$ and $CCN_{0.5}$) from Macquarie Island between April 2016 to March 2018. Seasonal cycles of $CN_{10}$ (b) and $CCN_{0.5}$ (c) from Macquarie Island (MQI, 2016-2018), Kennaook/Cape Grim (KCG, 2011 - 2020) and Syowa (SYO, 2004 - 2016) are shown, with monthly medians (solid line) and interquartile range (shaded regions) shown. Overlaid on the seasonal cycles are the weekly medians from ship-based campaigns coloured by latitudinal bins. Note the y scale of (c) has been adjusted to highlight the seasonal cycle, but this excludes a data point from MARCUS in March showing a weekly $CCN_{0.5}$ median concentration of $586 \, \mathrm{cm}^{-3}$ corresponding to the $CN_{10}$ data point of $897 \, \mathrm{cm}^{-3}$ in (b).

delayed by about 2 months. Overall there is good agreement between concentrations observed at SYO and MQI, with the exception of the high late summer concentrations observed at SYO, but absent at MQI.

Source of aerosols in this region are primarily wind-driven sea salt and secondary aerosol formation from the oxidation of

biological emissions. It is this latter source which is generally accepted to be the primary driver of the aerosol seasonal cycle in the Southern Ocean and Antarctic region. Around Antarctica in particular, the cold water and abundant marine nutrients serve as ideal breeding grounds for large communities of phytoplankton, which are known to reach their maximum abundance in late summer (Deppeler and Davidson, 2017). While still present at lower latitudes, these phytoplankton communities are less abundant. In addition, sea ice algae build up over winter at the higher latitudes, and release their emissions during the

melting season in spring and summer (e.g. Jang et al., 2022). These spatial and temporal differences between the Southern



Ocean stations of KCG and MQI, compared with the Antarctic station, go a long way to explain the differences in timing and magnitude of the summertime maxima.

Also at play here is the airmass transport pathways influencing the different stations. Previous studies (Humphries et al., 2016, 2021a; Alroe et al., 2020a; Simmons et al., 2021; Chambers et al., 2018) have established that a distinct atmospheric

compositional boundary exists, known as the Atmospheric Compositional Front of Antarctica (ACFA), that is typically found between 60°S and 65°S in the East Antarctic sea ice region. The ACFA has been found to reduce the flow of air between the higher and lower latitudes (Humphries et al., 2016). In particular, emissions from phytoplankton communities around the Antarctic coastline in summer tend to not be homogeneously mixed throughout the Southern Ocean atmosphere, and instead, influence the Antarctic continent more so, leading to the differences observed in aerosol concentrations between stations in this

study.

Further to the above, the delayed increase in concentrations observed at MQI relative to the other two stations is likely the result of the lower abundance of phytoplankton communities, and absence of sea ice algal communities, at the latitudes Macquarie Island is situated. Phytoplankton communities are found (Deppeler and Davidson, 2017) to congregate more around coastlines where upwelling of nutrients occurs more frequently than in the open ocean (i.e. at MQI latitudes). This results in

higher precursor emissions at these latitudes, leading to secondary particle formation and a rapid increase in concentrations at the stations closer to larger landmasses (i.e. KCG and SYO). It is plausible that the delayed increase in concentrations at MQI is a result of the delayed increase in phytoplankton populations in the open ocean together with the increased time required for cross-latitudinal airmass mixing to occur.

The wintertime minimum in SYO is substantially lower than those of the other two stations. In wintertime in the Southern

Ocean and Antarctic region, the biological driver of aerosols is largely absent, and aerosol concentrations are driven primarily by wind-driven sea salt. While all three stations are on the coastline surrounded by water, SYO is unique in that its surrounding ocean is covered in sea-ice all year-round, with its wintertime extent reaching approximately 1000 km. This results in a significant reduction in the ability for high winds to produce sea salt aerosol, and is likely the explanation of the difference observed between the stations during wintertime across the three stations.

The differences observed between MQI and KCG are worth noting. Both stations are in the Southern Ocean, however the $CN_{10}$ concentrations observed at KCG are higher thank those observed at MQI throughout the year (with the exception of midwinter), reaching up to double the concentrations during some periods. KCG has long been thought of as being representative of the Southern Ocean region in general. While true for many atmospheric species, a recent study analysing data from MARCUS and CAPRICORN2 (Humphries et al., 2021a) showed that summertime aerosol concentrations were substantially higher than

those at higher latitudes, with observations at KCG representative only of voyage observations north of 45°S. A voyage will take place in 2025, one goal of which is to experimentally answer the question of how representative observations at KCG are of the wider Southern Ocean. It may be the case that the region north of 45°S is somewhat distinct from the higher latitudes and the drivers of this difference should be investigated further.

Weekly voyage $CN_{10}$ medians are shown overlaid on station seasonal cycles in Figure 2b, and are coloured by latitudinal

bands. These bands have been chosen based on results of Humphries et al. (2021a) where the region exhibits different aerosol





populations across three primary sectors: north of 45°S; Southern Ocean mid-latitudes from 45°S to 65°S; and south of 65°S. Humphries et al. (2021a) established these divisions based on summertime voyage data split into 5° latitudinal bins, however it is known that the ACFA varies latitudinally with time, and its position is likely to vary significantly with season, similar to the oceanic polar front (Seidel et al., 2008; Lucas et al., 2014; Davis and Rosenlof, 2012; Choi et al., 2014). Unpublished

data from the summer CAPRICORN2 voyage show that the position can vary several degrees in a short time span and by longitude - the ACFA was observed at around (63°S, 140°E) and a week later at (65°S, 150°E). The springtime SIPEXII voyage observed the ACFA at (64.4°S, 140°E). Unfortunately, data outside of summer in this latitudinal band are very limited so thorough characterisation of the seasonal location is impossible with currently available data sets. This variability, together with the likely larger variability associated with seasonal changes, has led the authors to establish a fourth latitudinal band for

this analysis, from 60°S to 65°S, where data are available.

Overall, voyage data appear to agree well with long-term seasonal data sets. Since most of the voyages occurred south of Tasmania, only limited data are available for the northern sector (orange markers). These data all fall within the ranges of those observed at KCG. Similarly, voyage data from the mid-latitudes (blue markers) agree well with data measured at MQI, with the exception of a single week in March which was anomalously high, and was likely a result of measuring outflow from

Tasmania (this was measured during MARCUS approximately halfway between Hobart and MQI). Voyage data in the polar cell sector (green markers) agree well with SYO concentrations throughout the year, except for during the February maximum at SYO, where voyage data are below SYO's 25th percentile. This could be a result of longitudinal variability, since Syowa is a long distance west of where these voyage data were obtained. The transitional sector between 60°S and 65°S (cyan markers) is found to largely be on the lower end of the seasonal distributions, and is observed to typically be consistent with mid-latitude

Southern Ocean $CN_{10}$ data.

### 3.2    CCN seasonal cycles

The seasonal cycle of $CCN_{0.5}$ is shown in Figure 2c for platforms where those data are available (18 month version found in Appendix Figure A1b). Unfortunately, CCN data was unavailable at SYO. As expected, the overall pattern of the CCN seasonal cycle follows that of CN for each site, with a typical summer maximum and winter minimum. For both KCG and MQI where

seasonal CCN data are available, the summer maxima occurs in January, consistent with CN observations at the same sites. For KCG, CCN data reach their minimum in May, remaining at approximately the same minimum value until they start to rise in October. This is in contrast to the CN minimum, which only reaches its minimum in June and July. At KCG then, the increase in CCN over the warmer months occurs over a much narrower time period than does the CN - seven months above background compared with 10. It could be that this pattern is a result of insufficient precursor gases (i.e. oxidation products of biological

emissions such as DMS and MSA) in the shoulder seasons resulting in aerosol populations at sizes too small to be able to act as CCN. This is an interesting question and should be investigated further looking at multi-year time series of aerosol size distributions. While routine observations of aerosol size distributions have recently begun at KCG, data are not yet available for comprehensive analysis, and will be a topic of future work.




The most surprising characteristic of the available seasonal cycles is the significant spike in $CCN_{0.5}$ concentrations at MQI

in May and June. This feature, present at all supersaturations (Appendix Figure A2), is of the same magnitude as the summer maximum in CCN, but interestingly, is not one that is observed in the $CN_{10}$ data. This feature was observed during the first year of data (2016), and unfortunately, could not be confirmed whether it was a persistent feature in the second year of data because of instrument malfunction during this period. In addition, because of unfortunately timed malfunctions of the CPC, this feature could also not be investigated by interrogation of the CCN/CN activation ratio (Appendix Figure A3). Investigation

of what factors could be driving this peak show that there were no instrumental issues or local site activities that could be responsible. It could be possible that this feature is driven by a small number of data points biasing the statistics used to create the seasonal cycle, however Appendix Figure A4 shows that there was almost no downtime during this period in that first year, and that although only 50% of data are available over the entire campaign period (i.e. only one of the two years is available), this period isn't anomalously low such that poor statistics would explain this feature.

If a natural feature, it would suggest a change in the composition of the aerosol population during the period. To investigate this further, we probed into both the meteorological and radon data during this period to determine whether there were any obvious drivers. Unfortunately, there were no significant correlations of meteorological parameters with this feature, either positive or negative. It is plausible to suggest that these increases are a result of increased continental influence from long-range transport from Australia, although the absence of an increase in CN, and more importantly, radon (Appendix Figure A5),

would suggest this isn't the driver here. A closer look at the variability broken down by supersaturation (Figure A2) suggests the highest variability occurs in the lower supersaturations, suggesting this is being driven by sea salt aerosol. The existence of this spike warrants further multi-year observations at the station to understand the driving factors, given it is of similar magnitude as the summer maxima.

Weekly voyage $CCN_{0.5}$ medians are shown overlaid on station seasonal cycles in Figure 2c. Voyage CCN data are found

not to agree with station data as well as the $CN_{10}$ data. Voyage data from the northern sector (orange markers) agree well with those observed at KCG during summer months, but the limited data available in this sector in springtime falls well outside the typical range for KCG at this time of year. Mid-latitude voyage data (blue markers) agree well with each other, but don't seem to be consistent with data from MQI, tending to be lower in late autumn (noting that this is the timing of the anomalous winter peak at MQI), and higher in summer. Data in the transition zone (cyan markers) are consistent with those of mid-latitudes.

Notably, polar cell data (green markers) are markedly higher during the late summer months, which is consistent with late summer peak in $CN_{10}$ concentrations observed at SYO.

### 3.3    Seasonality of activation ratio

The CCN/CN activation ratio is an important parameter that helps to understand the population size distribution and composition. Activation ratios of unity can be interpreted as all available CN data can serve as CCN, meaning the accumulation and

larger Aitken modes are dominant, and species are typically water soluble. Lower activation ratios mean either the composition is largely organic species, or a strong nucleation or Aitken mode is present, or both. Seasonal cycles of the activation ratio, together with voyage data, are presented in Figure 3. At KCG, median ratios range from 0.2 in late spring, to 0.5 in winter.





Values appear to have three distinct periods: summer and autumn are reasonably constant, exhibiting values around 0.3; a winter maximum, with values reaching up to 0.5; and a spring minimum around 0.2. This is consistent with the seasonal cycle in aerosol populations being driven by secondary particle formation, resulting in a strong Aitken mode which doesn't all grow to CCN active sizes. In the winter, the source of secondary particles all but disappears, leaving predominantly sea salt aerosols, which tend to occur most prominently in the accumulation and coarse modes and can all act as CCN.

The seasonal pattern observed at MQI largely follow that of KCG, but with some interesting differences. Values are similar during summer and autumn. This is likely because the phytoplankton blooms have had sufficient time to ramp up, and subsequent precursors of the secondary aerosol have become thoroughly mixed through the Southern Ocean atmosphere, resulting in similar aerosol populations - albeit at different concentrations - being observed at both sites. Interestingly, winter and spring values at MQI are approximately 0.2 higher than those observed at KCG. Winter is when biological activity is at its minimum, and during springtime, blooms are only just beginning in geographically limited areas, particularly those around major landmasses. This results in KCG being more responsive to small concentrations and changes in phytoplankton abundance, resulting in aerosol populations in these seasons having a stronger Aitken mode (as evidenced also in the CN and CCN seasonal cycles in Figure 2) that reduces the activation ratio relative to MQI.

Voyage activation ratio data show mixed results. Spring ratios are low compared with their respective latitudinal station. During summer and autumn, some mid-latitude values agree within the range of those observed at MQI, but many points also seem to extend to 0.8, suggesting a weighting away from smaller diameter aerosol populations during these periods. Observations in the high latitudes in summer fall into the range of 0.6 to 0.8, similar to many of those classified as mid-latitude. This could suggest that many of these mid-latitude values are actually measuring air that has recently been in the Polar Cell. These very high activation ratios, during a period where secondary aerosol formation is dominant, suggest that these secondary aerosols are growing well above the required activation diameter. Given the abundance of phytoplankton at these latitudes, it is likely that precursor gases are present in sufficient concentrations to grow particles quickly to CCN active sizes. The fact that we don't see similar activation ratios at the northerly latitudes suggests a much weaker precursor source, resulting in insufficient condensational growth to CCN sizes.

### 3.4 Latitudinal changes as a function of season

The data set being utilised in this study contains a valuable amount of both spatial and temporal information. In Figures 4 and 5, population distributions are shown for $CN_{10}$ and $CCN_{0.5}$, respectively, split into seasons and previously described latitudinal bands (accompanying summary statistics are presented in Appendix Tables A1 and A2).

#### 3.4.1 $CN_{10}$

Summer $CN_{10}$ data (Figure 4a) confirm the latitudinal summertime patterns observed by Humphries et al. (2021a), where the highest concentrations are observed in the northern sector at latitudes lower than 45°S; the lowest concentrations are observed in the mid-latitudes of the Southern Ocean (45°S and 65°S), and a small increase in concentration is observed in the high latitude bin (> 65°S). Voyage and station data both agree well within each latitudinal band, except for the polar cell band.





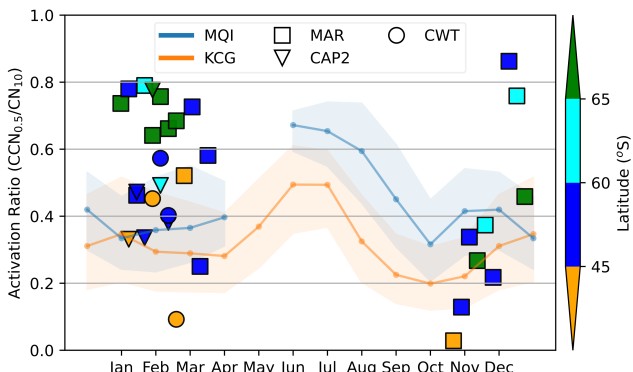

**Figure 3.** Seasonal cycle of the activation ratio, $CCN_{0.5}/CN_{10}$ at both Macquarie Island and Kennaook/Cape Grim, with monthly medians (solid line) and interquartile range (shaded regions) shown. Overlaid on seasonal cycles are weekly medians from voyages where data are available, as in Figure 2.

This is consistent with previous observations (Humphries et al., 2016, 2021a) which found that concentrations within the sea ice region around the Antarctic continent (termed the Antarctic Sea Ice Atmospheric Compositional Zone; ASIACZ) were found to exhibit distinctly different aerosol properties to observations on the continent itself. We note that this is the first time continental and ASIACZ data sets have been obtained simultaneously to be able to compare directly and suggests that the location of the ACFA during this period was somewhere between 65°S and 69°S (69°S being the latitude of SYO).

There are no voyage data in Autumn in the ACFA and ASIACZ latitudes (Figure 4b). Voyage data agrees well with station data at mid-latitudes, however this isn't the case at the northern latitudes where voyage data are significantly higher than station data from KCG. This is likely a result of the low sample size of voyage data in this sector, and also that voyage data are not filtered for baseline conditions (i.e. wind directions and radon concentrations) like KCG data are, resulting in a higher continental influence. Again, the northerly bin shows the highest concentrations, which decrease further south. During this season, SYO data are comparable with mid-latitude data, albeit with a slightly elevated upper 50% of data, likely driven by the comparatively higher March concentrations reflecting the later summertime peak discussed early (Figure 2b).

No voyage data exist during the winter months (Figure 4c), meaning we only have station data to understanding latitudinal differences, and only very limited information can be gained about the ASIACZ and ACFA region during this period. Results here reaffirm what was concluded from Figure 2b where populations observed at KCG and MQI are similar, reflecting the dominance of sea salt aerosol and their similar proximity to the open ocean. The slightly higher concentrations observed at MQI is likely the result of it being closer to the sea surface, it's inlet being only 10 m above sea level, compared with 94 m at KCG. SYO's concentrations are substantially lower (medians approximately half of northern stations) resulting from the significant isolation of the station from the open ocean (~1000 km) in winter.

We are fortunate to have a significant amount of voyage data available during the spring months (Figure 4d) - a result of SIPEXII directly targeting the mid-spring season, and MARCUS observations that began in mid-October inline with when resupply operations for Australia's Antarctic program become feasible. Springtime latitudinal patterns in general follow those





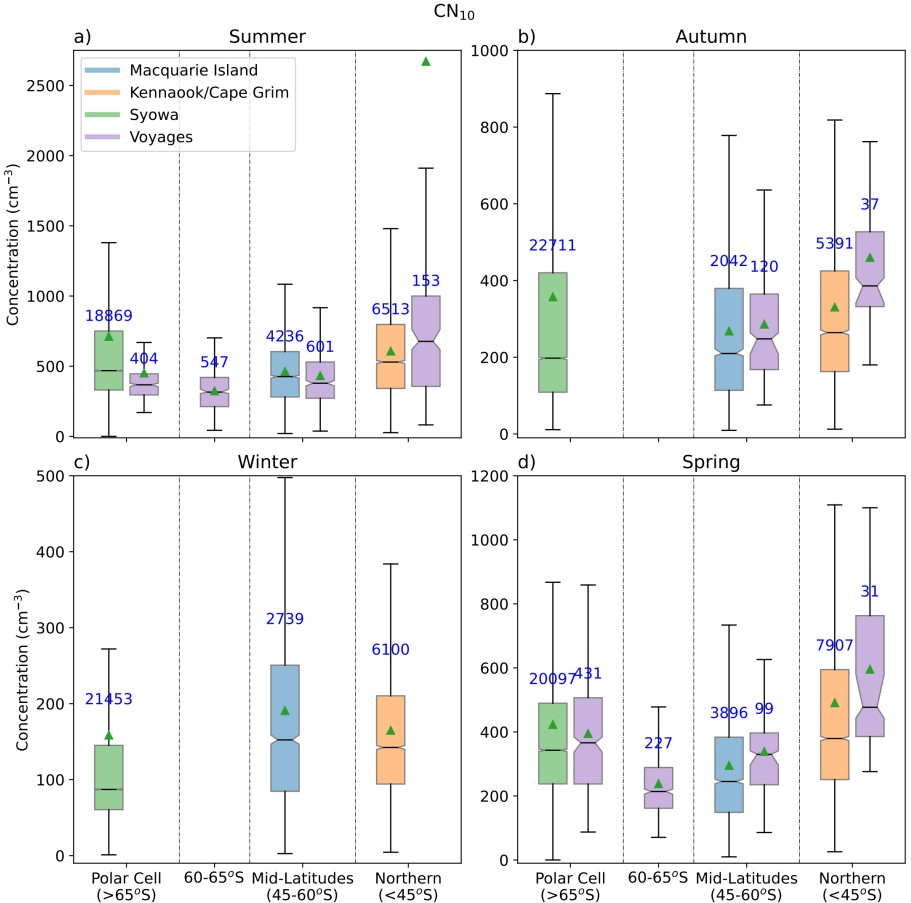

**Figure 4.** Box plots of $CN_{10}$ concentration categorised into seasons and latitudinal sectors in line with definitions by Humphries et al. (2021a). The 60-65°S sector is divided out as this is a boundary zone, the latitude of which varies significantly on daily timescales. All voyages are combined into a single data set (shown in purple). Box plots are standard Tukey plots with the boxes representing the 25th, 50th and 75th quartiles, and the upper (lower) whisker being the first data points above (below) the 75th (25th) quartile plus (minus) 1.5 times the interquartile range. The notch in the box represents the confidence interval of the median, as calculated from 1000 bootstrap iterations. Green triangles represent the mean of each data set and the numbers above each box show the number of hourly observations used to calculate each box plot. A similar figure is shown in Figure A6 with all y axes limits kept the same.

of summer, with northern concentrations being the highest, followed by those at the highest latitudes, and the lowest being in the mid-latitudes. Unlike summertime concentrations where data from SYO and voyages in the polar cell were significantly different (medians of 468 $cm^{-3}$ and 368 $cm^{-3}$, respectively), observations in springtime are almost identical (343 $cm^{-3}$ and 366 $cm^{-3}$, respectively). It is worth noting here that while there is good agreement between springtime $CN_{10}$ concentrations, previous analysis (Humphries et al., 2016) of SIPEXII data and comparison with the literature show that there is significant differences if $CN_3$ concentrations are compared, suggesting a strong source of recently nucleated particles in the ASIACZ that





is absent in coastal regions. It is suggested that observations of $CN_3$ are an important enhancement to measurement programs

in this region in order to understand the drivers of this change (noting that this parameter is available for KCG and many of the voyages utilised in this study). The difference between concentrations observed in the polar cell, with those in the 60-65°S latitudinal band is also striking, and would suggest that in springtime, this band is actually more reflective of mid-latitude marine aerosol populations. Both of these features (i.e. the differences in the polar cell band between spring and summer, and the springtime difference between the polar cell and the 60-65°S bin) together suggest the location of the ACFA in springtime

is much farther north than in summer. This is consistent with the understanding that the position of the ACFA is driven largely by meteorology, and is a somewhat real-time realisation of the forces that create the atmospheric polar front (Humphries et al., 2016).

### 3.4.2   CCN

When looking at $CCN_{0.5}$ spatial distributions (Figure 5), we are unfortunately missing valuable data from SYO to give us

insight into the differences between the ASIACZ and coastal Antarctica. Analyses of available data show that summertime CCN (Figure 5a) are again consistent with spatial patterns derived from previous publications (Humphries et al., 2021a), with the highest concentrations observed in the northern sector, followed by the lowest concentration in the mid-latitudes, and a substantial increase in the polar cell bin south of 65°S. Concentrations in the 60-65°S sector fall within those expected at mid-latitudes during the summer period. Interestingly, voyage data in the mid-latitudes are substantially higher than those observed

at MQI or in 60-65°S sector, and is actually more in line with populations observed in the northern sector. This is the case for all voyages where data are available (MARCUS, CAPRICORN2, Cold Water Trial and PCAN), however MARCUS is distinctly higher (not shown). Since this is not apparent in the $CN_{10}$ data (Figure 4a), it would suggest that a big difference in the latitudinal patterns that we observe is actually in the size distribution - where the northern sector has a strong and persistent Aitken mode that is significantly weaker in more southerly locations. Further analysis of aerosol size distributions, decomposed

into finer scale (e.g. 5°) latitudinal bands, would help shed light on these changes. These data are available from the RVI-GAW station for many years, and will be a topic of a future publication.

Autumn voyage data (Figure 5b) are consistent with the expected pattern, however concentrations in the northerly sector are drastically higher than those observed at KCG. This is likely a result of the small sample size (only 77 hourly points) that is biased towards the early autumn months when many of these voyages were ending their summer campaigns. Given

the significant changes that occur in autumn, the timing of the sampling has a big influence on the data. Data from MQI are substantially higher than at KCG (a feature that is also observed in winter data in Figure 5c), and are also higher than voyages in the mid-latitude sector. This is driven by the unexplained late-autumn/winter peak in CCN observed at MQI previously discussed.

The limited voyage $CCN_{0.5}$ data during springtime shows significantly lower concentrations in the north compared to those

south of 60°S. Although the number of hourly samples available from voyages during this time of year is limited, this change could be explained by the rapid increase in phytoplankton and sea ice algae that is concentrated more at higher latitudes than in the north. A word of caution is warranted here though, the low sample number could be significantly biasing the results,





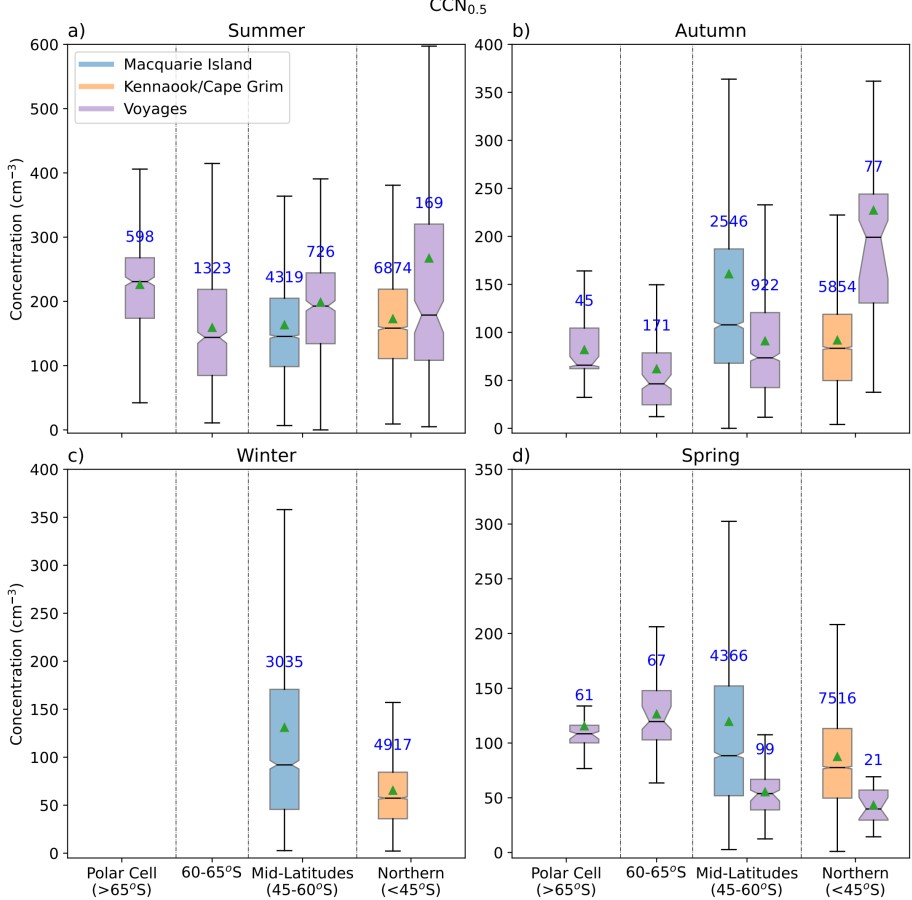

**Figure 5.** As in previous figure, but for $CCN_{0.5}$. A similar figure is shown in Figure A7 with all y axes limits kept the same.

particularly given that springtime is known to be a period of significant change in the region, meaning that a small sample size at one particular period, is not reflective of the entire season. This is likely the reason that voyage data does not compare well

with station data during both springtime and autumn. This is a key point that reiterates the importance of long-term sampling, rather than just relying on campaign observations, particularly in shoulder seasons.

Overall, it can be concluded from these data that long-term stations are good representations of their respective latitudinal bands. Although we are fortunate to have two years of recent data from MQI, the establishment of long-term observations at this location is imperative to monitor the changing climate in this sensitive region. The infrastructure for long-term observations

already exists (e.g. Stavert et al., 2019), meaning that it is just the funding that is required. SYO's aerosol program is invaluable and would be significantly enhanced by the addition of routine CCN and $CN_3$ observations.

Long-term observations within the atmospherically distinct, and spatially varying ASIACZ is an important missing part of the puzzle, resulting in a distinct absence or sparsity of data in shoulder and winter seasons. Long-term observations in this region would be logistically very challenging given the dynamic nature of the pack-ice environment. An ideal way to improve





these observations is to establish long-term measurements programs aboard ice-breaking vessels frequenting the Antarctic
continent. This would establish frequent, repeated observations of the region, and importantly crossing of the ACFA (thereby
improving the characterisation of its varying location). The primary limiting factor though, is that observations outside of resup-
ply seasons does not occur, resulting in no observations in winter, and possibly autumn (depending on the vessel). Consideration
of low-power, unattended buoys outfitted with suitable instrumentation for the observations of atmospheric composition may
also be possible as technology improves. Logistically, long-term winter observations may be out of reach at this point in time,
however there is already discussion in the community about undertaking a year-long campaign in the Antarctic region, similar
to the MOSAiC campaign (Shupe et al., 2020) undertaken in the Arctic 2019-2020. Such a campaign would be a step change
in Antarctic science across a range of disciplines that are limited by similar seasonal constraints.

## 4    Conclusions

Observations of $CN_{10}$ and CCN from a two-year campaign at Macquarie Island (MQI; 58.5°S, 158.9°E) between March 2016
and March 2018 are presented for the first time. Seasonal cycles show low concentrations typical of other sites in the region,
with summer maxima and winter minima, driven primarily by biological emissions. Seasonal cycles were compared with two
other long-term stations in the region: Kennaook/Cape Grim, Australia (KCG; 40.7°S, 144.7°E), and Syowa station, Antarctica
(SYO; 69.0°S, 39.6°E).

The timing of the seasonal features in $CN_{10}$ differs between each of the long-term stations. Summer maxima occurred in
January at the Southern Ocean sites, but was delayed by a month at SYO. Autumn decreases occur rapidly at SYO and MQI,
with a more gradual decline at KCG. Winter minima lasted the longest at MQI (May through September), followed by SYO
(May to August) and KCG having the shortest low (June to August). Aerosol populations at both KCG and SYO were quick
to respond to the start of spring in September, however concentrations at MQI only began ramping up strongly in November.
Most of these features could be explained by the spatial and seasonal changes in phytoplankton communities across the region.

For CCN, data was unavailable from SYO, so a detailed latitudinal understanding could not be obtained. However, the
differences at KCG between the CN and CCN are worth noting - the former's summertime maximum enduring much longer
(10 months) than does the latter (7 months). At MQI, the summertime maximum of $CCN_{0.5}$ is not as strong as is observed
at KCG, however a peculiar increase in CCN concentration across all supersaturations is observed in May and June of 2016.
Unfortunately, instrument failures at this time of year in the second year of observations prevented confirmation that this was
a persistent feature, or an anomaly occurring in just that year. This increase did not correlate with radon or any meteorological
parameters.

Activation ratio data showed winter maxima reflective of the absence of secondary aerosols that drive this ratio down, with
primarily only sea salt aerosol remaining. Activation ratios at MQI in this period were 0.2 higher than those at KCG, likely
the result of the station being further away from biological sources still active in the winter. Summer observations in the high
latitudes observed very high activation ratios up to 0.8. Given the very high abundance of phytoplankton communities in this
region, it is likely that precursor concentrations are strong enough to grow secondary aerosol populations to CCN active sizes.



Given the outflow of air masses across the Southern Ocean from this region, this is an important source of CCN to the Southern Ocean region as a whole.

Analysis of how latitudinal gradients change with seasons show that summertime CN and CCN data all show the highest concentrations in the northern sector, low concentrations in the mid southern ocean latitudes, followed by a modest increase in the polar sector. This pattern is also consistent for shoulder seasons, however the winter pattern differs significantly, with consistent concentrations at both mid and northern latitude sites, and a significant decrease at SYO. Detailed latitudinal breakdown for seasons outside of summer was not possible because of the significant dearth of voyage data. The location of the ACFA

was found to vary consistent with seasonal expectations, where springtime position is further north compared to springtime observations. Again, characterising its location in autumn and winter seasons is not possible because of the lack of ship data during these periods.

While this two year data set at Macquarie Island is an important step forward in our understanding of the seasonal cycle in this region, it was limited both in duration and the types of instrumentation. In addition to the establishment of new long-

term monitoring stations at both mid Southern Ocean latitudes, within the Antarctic sea ice region, and on the Antarctic continent, these studies repeatedly highlight the requirement for enhancing all long-term station observations to include at a minimum, $CN_3$, $CN_{10}$, CCN and aerosol size distributions. Instrumentation for these measurements is mature enough to be largely automated and run with only minimal on-the-ground technical support together with remote access. The addition of composition information is highly desired information, but is not as automated and can be more expensive.

*Data availability.* All data utilised in this work are publicly available. CN and CCN data from Macquarie Island are available at https://doi.org/10.25919/g7jx-k629 (Humphries et al., 2021c). Data from Kennaook/Cape Grim are available at the World Data Centre for Aerosols at https://ebas-data.nilu.no. Data from Syowa are available at https://scidbase.nipr.ac.jp/modules/metadata/index.php?content_id=399&ml_lang=en (Hara et al., 2022). Data from MARCUS are available at https://doi.org/10.25919/ezp0-em87 (Humphries, 2020). Data from SIPEXII is available from https://doi.org/10.4225/15/5342423241BE4 (Humphries et al., 2014). Cold Water Trial data are available at https://doi.org/

10.25919/ytsw-9610 (Humphries et al., 2022b). CAPRICORN1 data are available at https://doi.org/10.25919/5f688fcc97166 (Protat and Humphries, 2020). Ice2Equator data are available at https://doi.org/10.25919/g07r-b187 (Humphries et al., 2022a). PCAN data are available at https://doi.org/10.25919/xs0b-an24 (Humphries et al., 2020). CAPRICORN2 data are available at https://doi.org/10.25919/2h1c-t753 (Humphries et al., 2021b).

**Appendix A: Appendices**

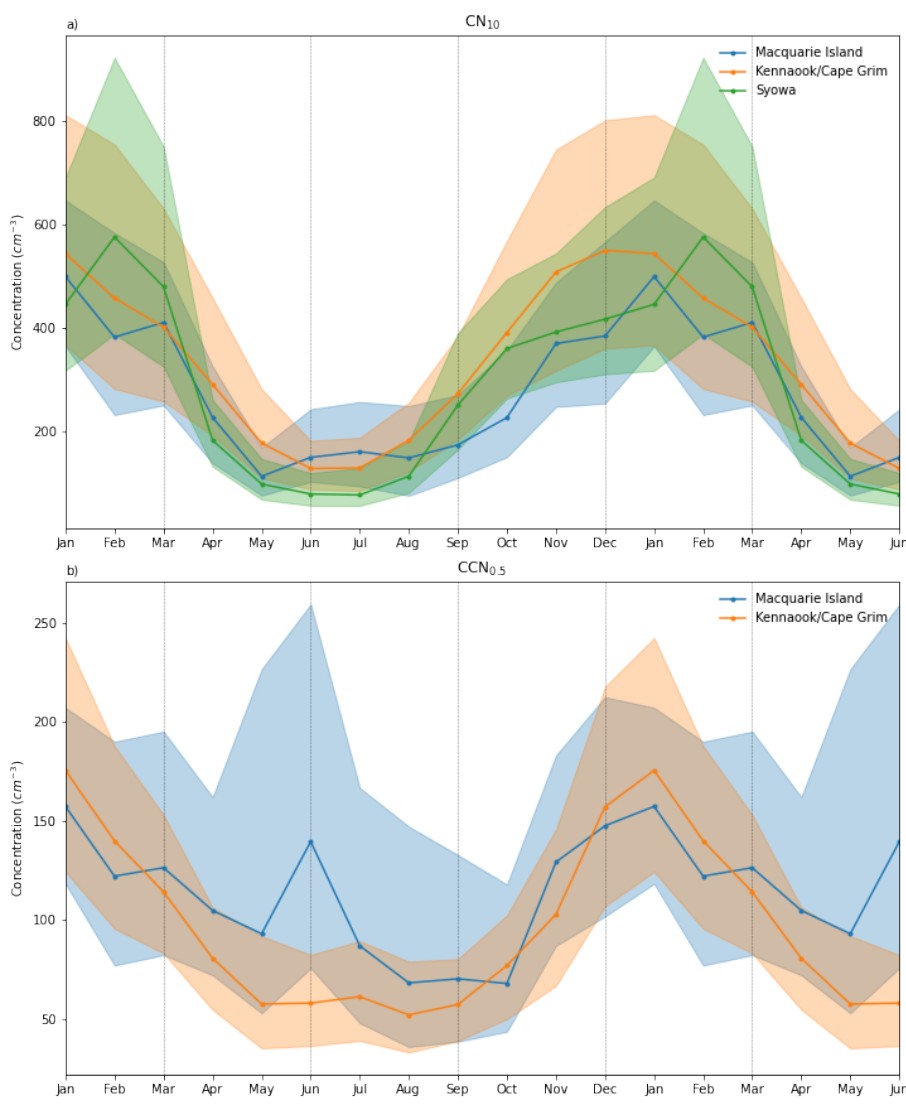

**Figure A1.** Annual cycles of $CN_{10}$ (a) and $CCN_{0.5}$ (b) at Macquarie Island (blue), Kennaook/Cape Grim (orange) and Syowa (green). The first six months are repeated to show the seasonal cycle over 18 months to enable visibility of both summer maxima and winter minima.



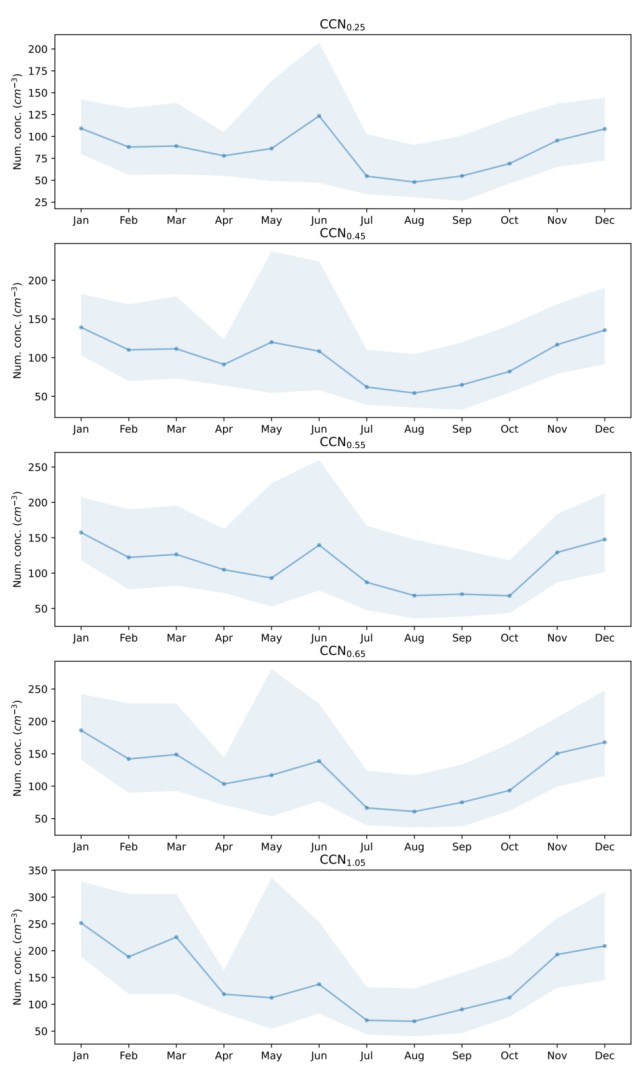

**Figure A2.** Annual cycles of $CCN_{0.5}$ at Macquarie Island for all five supersaturations measured.



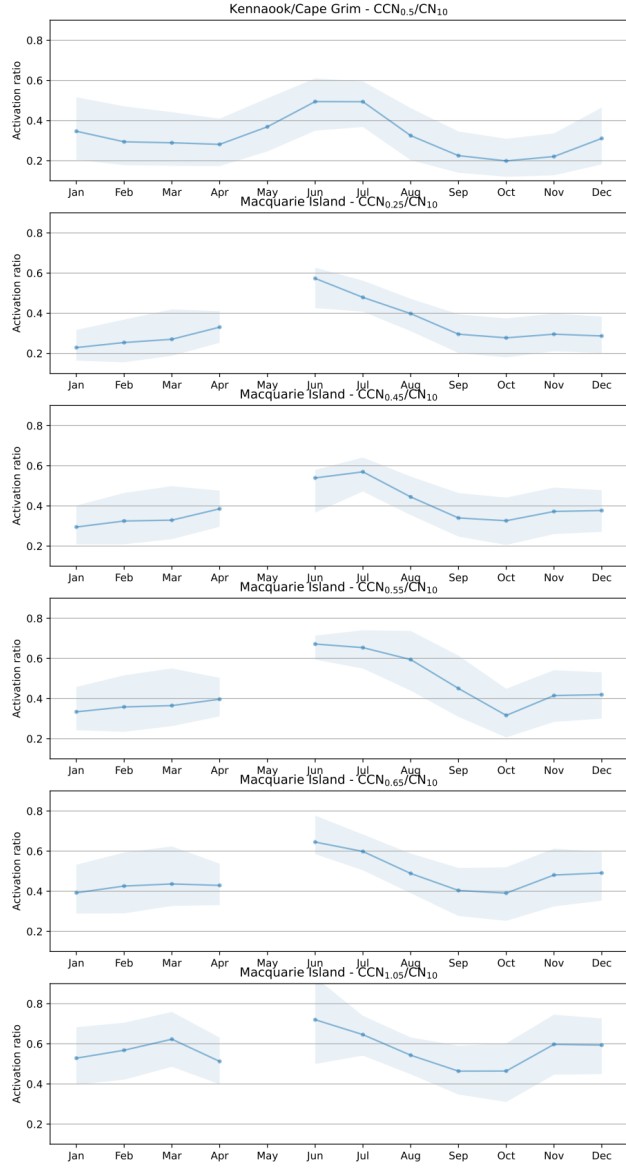

**Figure A3.** Annual cycles of the CCN activation ratio for Kennaook/Cape Grim (top) and then all observed supersaturations at Macquarie Island. Unfortunately, instrument malfunctions of either the CPC or CCNC resulted in no ratio data available for May and the first half of June at Macquarie Island.



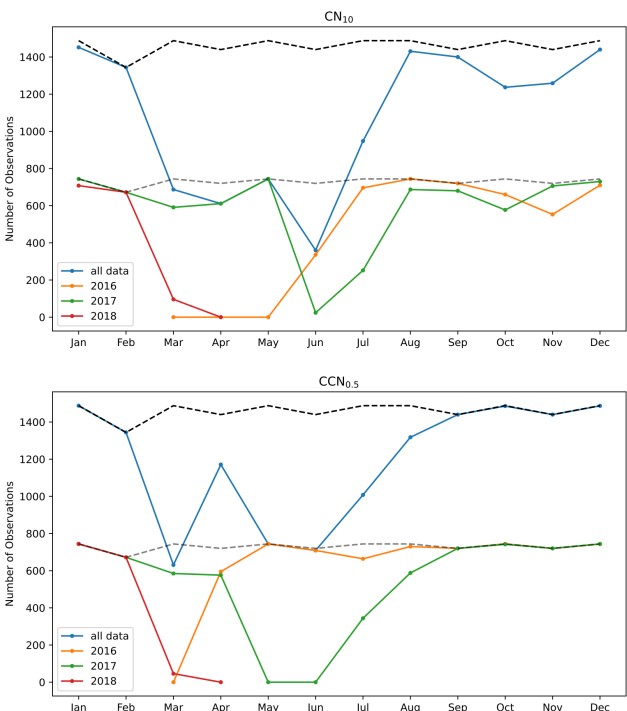

**Figure A4.** Number of hourly data points available per month from Macquarie Island during the two year for both $CN_{10}$ (a) and $CCN_{0.5}$ (b). Data are shown for the full campaign, and then separately for each year. The grey dashed line is a reference point for the maximum number of hours available in a particular month of each year, while the black dashed line is similar, but for the maximum available during a particular month over the entirety of the campaign.

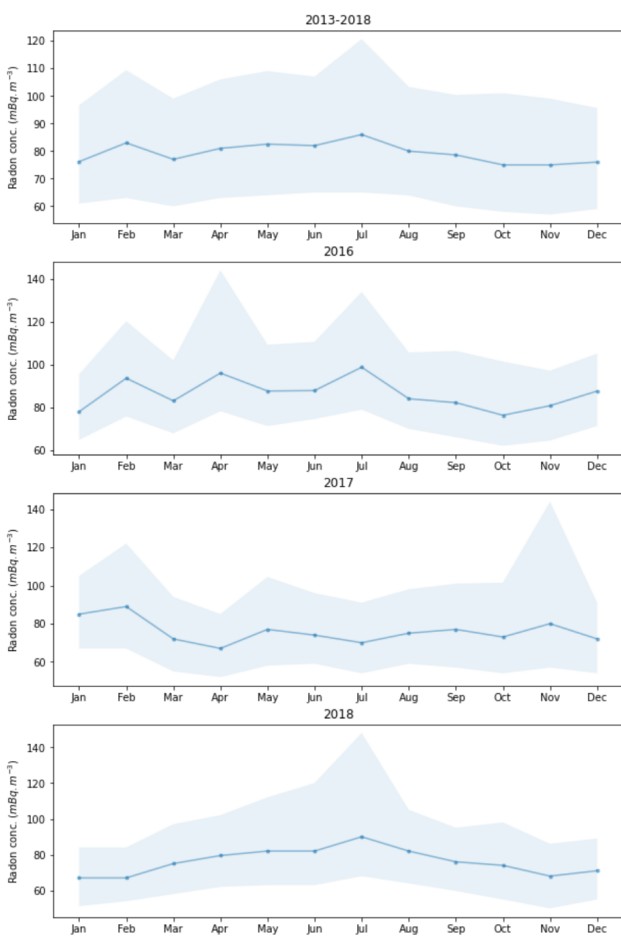

**Figure A5.** Annual cycles of radon for the six years of data available (top) and each separate year of the ACRE campaign. Note 2016 is where the CCN anomaly is observed.





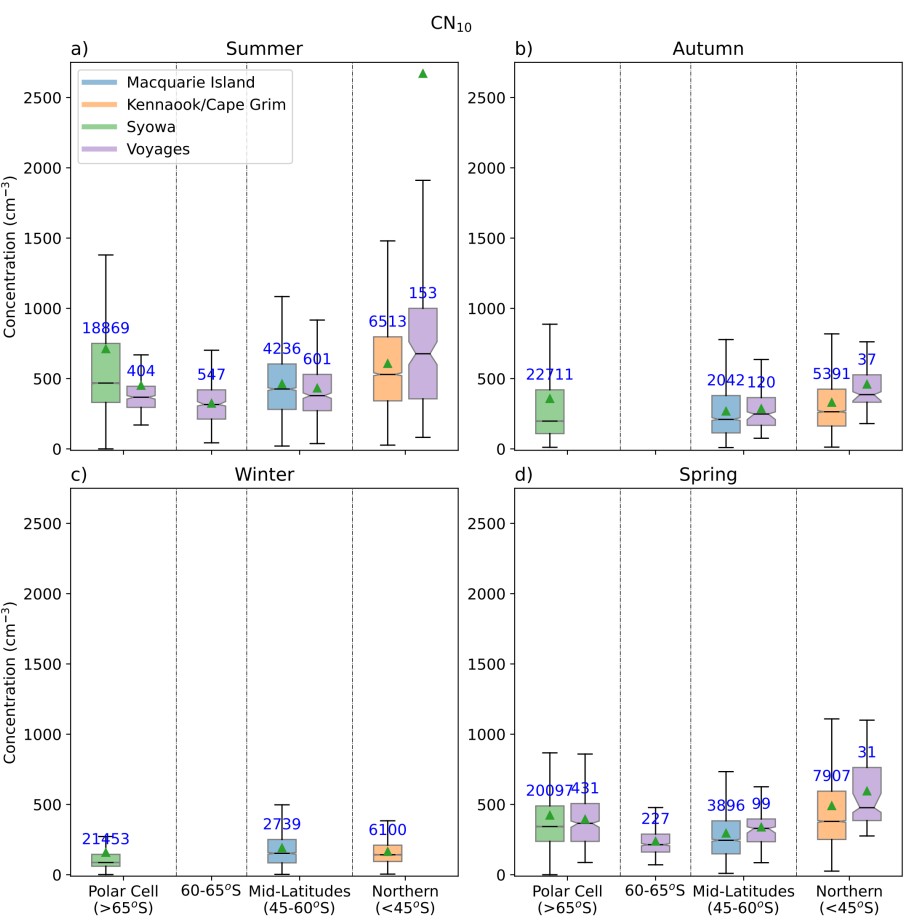

**Figure A6.** As in Figure 4 but with y axes limits kept the same across all seasons.

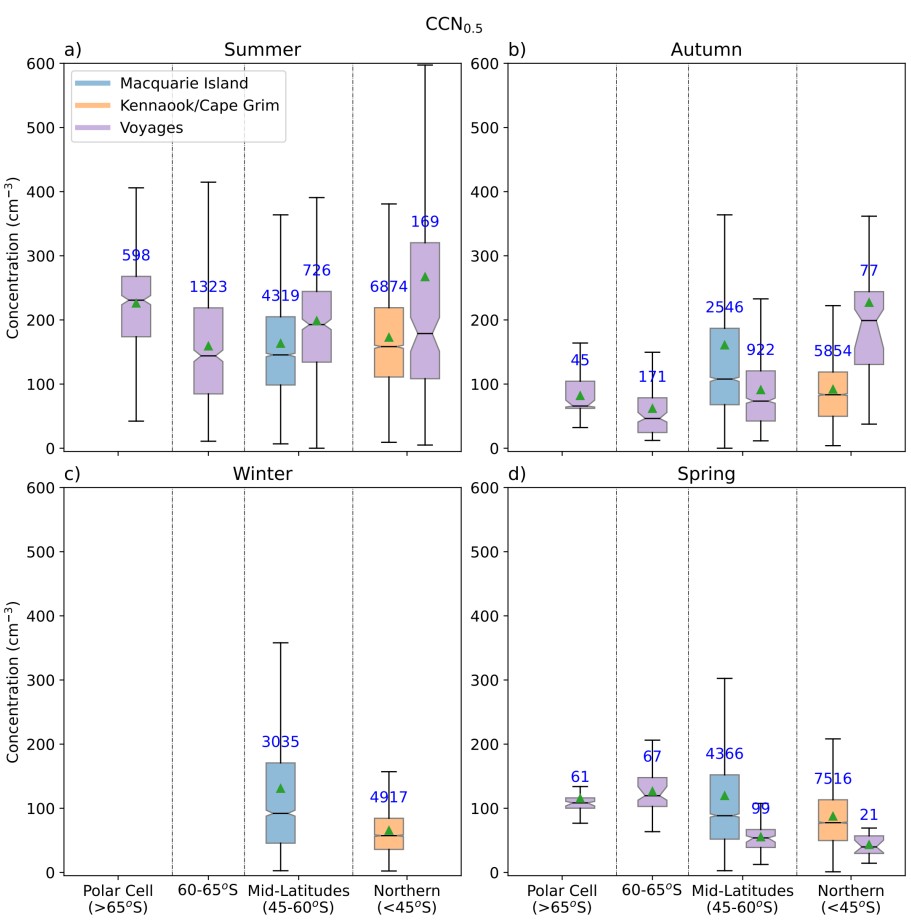

**Figure A7.** As in Figure 5 but with y axes limits kept the same across all seasons.





**Table A1.** Seasonal summary statistics from stations and voyages for $CN_{10}$

| Season | Statistic | Syowa | Macquarie Island | Kennaook/Cape Grim | Voyages | | | |
| --- | --- | --- | --- | --- | --- | --- | --- | --- |
| | | | | | >65 | 65°S-60°S | 60°S-45°S | <45°S |
| Summer | Mean | 712 | 462 | 607 | 451 | 324 | 432 | 2671 |
| | Median | 468 | 426 | 530 | 368 | 316 | 379 | 677 |
| | 25th | 331 | 281 | 342 | 295 | 212 | 272 | 356 |
| | 75th | 751 | 604 | 797 | 446 | 420 | 530 | 1000 |
| Autumn | Mean | 357 | 268 | 331 | - | - | 286 | 460 |
| | Median | 198 | 210 | 264 | - | - | 248 | 386 |
| | 25th | 109 | 114 | 163 | - | - | 168 | 332 |
| | 75th | 420 | 380 | 425 | - | - | 365 | 527 |
| Winter | Mean | 159 | 191 | 165 | - | - | - | - |
| | Median | 87 | 152 | 142 | - | - | - | - |
| | 25th | 61 | 84 | 94 | - | - | - | - |
| | 75th | 145 | 251 | 210 | - | - | - | - |
| Spring | Mean | 423 | 295 | 491 | 395 | 238 | 339 | 495 |
| | Median | 343 | 245 | 380 | 366 | 214 | 330 | 477 |
| | 25th | 238 | 149 | 251 | 237 | 162 | 235 | 386 |
| | 75th | 490 | 383 | 595 | 506 | 289 | 397 | 763 |





**Table A2.** Seasonal summary statistics from stations and voyages for $CCN_{0.5}$

| Season | Statistic | Macquarie Island | Kennaook/Cape Grim | Voyages | | | |
|---|---|---|---|---|---|---|---|
| | | | | >65 | 65°S-60°S | 60°S-45°S | <45°S |
| Summer | Mean | 163 | 173 | 226 | 159 | 199 | 267 |
| | Median | 145 | 158 | 230 | 144 | 193 | 179 |
| | 25th | 99 | 111 | 174 | 85 | 134 | 108 |
| | 75th | 205 | 219 | 268 | 219 | 244 | 320 |
| Autumn | Mean | 161 | 92 | 82 | 62 | 91 | 277 |
| | Median | 108 | 83 | 66 | 46 | 73 | 199 |
| | 25th | 68 | 50 | 62 | 25 | 42 | 131 |
| | 75th | 187 | 119 | 104 | 79 | 121 | 244 |
| Winter | Mean | 131 | 65 | - | - | - | - |
| | Median | 92 | 57 | - | - | - | - |
| | 25th | 46 | 36 | - | - | - | - |
| | 75th | 171 | 84 | - | - | - | - |
| Spring | Mean | 120 | 88 | 116 | 126 | 55 | 43 |
| | Median | 88 | 78 | 108 | 120 | 54 | 40 |
| | 25th | 52 | 50 | 100 | 103 | 39 | 30 |
| | 75th | 152 | 113 | 116 | 148 | 67 | 57 |





*Author contributions.* R.H. wrote the manuscript and led the overall data analysis and interpretation. R.H. undertook the aerosol study design, deployment, instrument maintenance and data handling for observations at Macquarie Island and during CAPRICORN1, CAPRICORN2, Ice2Equator, PCAN and SIPEX2. M.K. is the lead scientist of the aerosol program at Kennaook/Cape Grim, and together with J.W. and J.H., maintain the instruments, sampling infrastructure and data quality control. S.A and A.K. were lead proponents of ACRE, MICRE and MARCUS campaigns, and instrumental in their field deployment. K.H. is lead scientist of aerosol observations at Syowa Station. I.M., R.H., J.W., and J.H. maintain the instrumentation and data production from aerosol instrumentation aboard the RV Investigator. A.P. was the chief scientist enabling observations during CAPRICORN1, CAPRICORN2, and Ice2Equator. J.A., L.C., B.M. and Z.R. maintained instrumentation and quality control of data for CAPRICORN1, CAPRICORN2, the Cold Water Trial and Ice2Equator. R.S. was chief scientist of the SIPEXII voyage and together with S.W. was instrumental in ensuring high quality measurements and in the data analysis. C.F. and G.R.K were PIs of the MARCUS data. G.M. was a chief scientist of CAPRICORN1 and CAPRICORN2 and was instrumental in undertaking analysis of these data, as well as MARCUS and ACRE data. G.M. was chief scientist of the MARCUS campaign. S.C., A.W., and A.G. are the lead scientists of the radon program at Macquarie Island.

*Competing interests.* The authors declare that they have no conflict of interest.

*Acknowledgements.* Technical and logistical support for the deployment to Macquarie Island were provided by the Australian Antarctic Division through Australian Antarctic Science Project 4292, and we thank John French, Peter de Vries, Terry Egan, Nick Cartwright, Ken Barrett, George Brettingham-Moore and Emry William Crocker for their assistance.

Funding for the ACRE and MICRE projects was provided by Australian Antarctic Science projects 4292 and 4431, and the United States Department of Energy through Grant DE-SC0018626. Funding for voyages was provided by the Australian Government and the U.S. Department of Energy.

MARCUS data were obtained from the Atmospheric Radiation Measurement (ARM) Program sponsored by the U.S. Department of Energy, Office of Science, Office of Biological and Environmental Research, and Climate and Environmental Sciences Division. We thank all the ARM technicians who collected the radiosonde and other data onboard R/V Aurora Australis. Technical, logistical, and ship support for MARCUS were provided by the Australian Antarctic Division through Australia Antarctic Science projects 4292 and 4387 and we thank Steven Whiteside, Lloyd Symonds, Rick van den Enden, Peter de Vries, Chris Young and Chris Richards for assistance. The SIPEXII project was funded by the Australian Antarctic Science Grant Program (AAS Project 4032) and several RV Investigator voyages were supported by Australia Research Council Linkage Infrastructure, Equipment and Facilities grant LE150100048.

The Authors wish to thank the CSIRO Marine National Facility (MNF) for their support in the form of sea time on RV Investigator and associated support personnel, scientific equipment and data management. In particular we thank the Seagoing Instrumentation Team, the Data Acquisition and Processing Team, the Data Centre and the MNF Operations Team for their technical, IT and logistical support.

The authors would also like to acknowledge the Australian Bureau of Meteorology for their long term and continued support of the Kennaook/Cape Grim Baseline Air Pollution Monitoring Station, and all the staff from the Bureau of Meteorology and CSIRO.

This research was supported in part by BER Award DE-SC0018995 (GM and RH) and NASA grants 80NSSC19K1251 (GM). The work of GMM was funded by the United States Department of Energy Awards DE-SC0018626 and DE-SC0021159. GK acknowledges support



from the Office of Science of the U.S. Department of Energy (DOE) as part of the Atmospheric System Research Program. The work of AP was partly funded by the National Environmental Science Program (NESP), Australia.

630    This project received grant funding from the Australian Government as part of the Antarctic Science Collaboration Initiative program (RH, MK, AP, SA, AK).

All data and samples acquired on the voyages are made publicly available in accordance with MNF and AAD Policies. Raw data from MARCUS (Kuang et al., 2018; Kulkarni et al., 2018) are available from the DOE ARM's Data Discovery at https://adc.arm.gov/discovery/, Humphries et al. (2021d) and Humphries (2020).





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
