# Peer review of "Measurement Report: Understanding the seasonal cycle of Southern Ocean aerosols"

_Atmospheric Chemistry and Physics, 2022_

## Referee Comment (RC2)

The study by Humphries et al. (2022) reports two years of recently acquired CN and CCN measurements on Macquarie Island, located in the mid-latitudes of the Southern Ocean. While this data set is still too short to study certain aspects and/or validate certain hypotheses, and the absence of size distribution data is also a limitation, these observations are very valuable as they contribute to characterise the aerosol in a region of the globe which remains poorly documented. In particular, as the authors point out, the possibility of documenting the seasonal variation of the concentrations is a real plus compared to campaign data, which only concern short periods of time and often target the same seasons; the combination with the other data sets explored also makes it possible to document the spatial variability of the particle concentrations in this region, which is particularly sensitive to natural (and in particular marine) aerosols and representative of pre-industrial conditions. I therefore recommend the publication of this paper, which is otherwise well written. I would, however, suggest restructuring section 3 to improve clarity. In particular, I would recommend merging sections 3.1 and 3.4.1 on the one hand, and 3.2 and 3.4.2 on the other hand, as they deal with the same topic, while keeping all the figures since they provide a complementary view. This would make it possible to group the information together and avoid repetition as well as seemingly contradictory conclusions, such as that in L372-373 ("These data all fall within the ranges of those observed at KCG.") and 466-468 ("Voyage data agrees well with station data at mid-latitudes, however this isn't the case at the northern latitudes where voyage data are significantly higher than station data from KCG.")

I also have some specific comments listed below (authors are invited to reflect the proposed suggestions/comments on the abstract and conclusion where appropriate).

P4, Methods: the data sets are well described but because of their number and the different treatments applied to "clean" them, it remains difficult in the end to get a clear overview of data availability for the different sites/campaigns. Since the Macquarie Island data are central to this study, I would suggest at least moving Figure A4 into the main text, to illustrate the explanations in Sections 2.1.1 and 2.1.2.

P6, L150-151 : « hourly statistics were calculated for each supersaturation": to clarify, as a result of the measurement sequence and data filtration, the hourly statistics are calculated for each supersaturation on the basis of 7 min of measurement (from 2/11/2016), is that right?

P7-8, L195-198: can the authors give a brief indication of the frequency of occurrence of data from the baseline sector? This would help understanding why the amount of data finally involved in the statistics (as indicated in figures 4-5 and A6-A7) is so limited in relation to the length of the period studied (11 years).

P8, L200-201: « The station has a significant sea ice presence around the station": this part of the sentence should be reworded.

P10, L271: isn't it after CAPRICORN1 voyage?

P11, Results and Discussion, general comments:

- I would recommend balancing the interpretation of certain observations by taking into account the variability associated with them or the magnitude of the differences observed; in particular, the repeated use of the terms "substantially" and "significantly" seems to me to be sometimes inappropriate (e.g. L344, L467, L504, L516, L519);
- Unless I am mistaken, the authors do not discuss the possible impact of 1) the difference in length of the datasets of the different stations and 2) the time lags between some datasets (e.g. SIPEXII in 2012, Cold Water Trial in 2015 vs. MQI in 2016-2018) on the observations, and, in relation to these aspects, a possible impact of e.g. interannual variability? Can the authors comment on this point?

P11, L309: « Note that the seasonal cycle is plotted over 18 months to show the maxima and minima clearly, is presented in Appendix Figure A1a": extra "is" should be removed.

P11, L310: « with the highest concentrations observed at the northernmost station, KCG. »: I do not fully agree with this statement since while the median CN concentration at KCG is often above that of other sites, this is not the case in February, when the concentrations at SYO are higher (visible on both the median and quartiles). I would simply suggest to slightly modify the sentence: « with the highest concentrations often / most of the time observed at the northernmost station, KCG ».

P11, L315-316: For KCG and SYO: looking at Figure 2.b, I would say that the concentration increase is already initiated in August (rather than September, as indicated in the text); this is especially the case for KCG, and seems more consistent with L387 (« This is in contrast to the CN minimum, which only reaches its minimum in June and July. »).

P11, L315-316: « Springtime increases begin almost simultaneously in September at both KCG and SYO, although the increase at MQI is comparatively": the connection between the two parts of the sentence does not seem appropriate.

P13, L348-349: « and is likely the explanation of the difference observed between the stations during wintertime across the three stations": this part of the sentence should be reworded.

P11-14 : Although explanations are given by the authors on the differences observed between the three sites, in particular in relation to their geographical position and configuration (small island vs. more extensive coastline), the singularity of the peak observed in February at SYO and the fact that these high concentrations are not found during the campaigns makes me curious: is this peak observed every year with a comparable amplitude, or are the observations in figure 2.a largely influenced by one or several years in particular?

P15, L394-395 : « the significant spike in CCN0.5 concentrations at MQI in May and June": from figure 2.c, I would say that the CCN0.5 peak is only seen in June; similarly, with the exception of CCN0.45, the increase in concentration for the other supersaturations is not striking in May in Figure A2, but again more marked in June.

P15, L424-426: « Activation ratios of unity can be interpreted as all available CN data can serve as CCN, meaning the accumulation and larger Aitken modes are dominant, and species are typically water soluble. Lower activation ratios mean either the composition is largely organic species, or a strong nucleation or Aitken mode is present, or both. ». This formulation seems to me to be oversimplified / incomplete since, beyond the intrinsic properties of the particles (size, chemical composition), supersaturation is a parameter to be taken into account in the capacity of the particles to serve as CCN: for a high supersaturation, it is not excluded that a dominant Aitken mode may be the source of a significant number of CCN.

P16, L445: « Observations in the high latitudes in summer fall into the range of 0.6 to 0.8, similar to many of those classified as mid-latitude.": it is not just 4 out of 10 points? Similarly, the use of "many" at L443 seems to me to be too strong.

P16, L446: couldn't this hypothesis be tested by an air mass back-trajectory study? (Similarly at L410, couldn't the possibility of long-distance transport from Australia be assessed by checking air mass back-trajectories?)

P17, L461-463 : « This is consistent with previous observations (Humphries et al., 2016, 2021a) which found that concentrations within the sea ice region around the Antarctic continent (…) were found to exhibit distinctly different aerosol properties to observations on the continent itself." : wording should be checked.

P19, L512-513: «concentrations in the northerly sector are drastically higher than those observed at KCG. This is likely a result of the small sample size (only 77 hourly points) ». There is a difference in the 75th percentile concentrations in summer of the same order as that observed in autumn (219 vs. 320 cm-3 in summer, 119 vs. 244 cm-3 in autumn) which is not discussed. Is the size of the campaign dataset (although larger than in autumn) suspected in this case too of explaining at least part of these observations?

P20, L527-528: « Overall, it can be concluded from these data that long-term stations are good representations of their respective latitudinal bands. » This conclusion does not seem to me to reflect previous discussions, in which notable differences between station observations and campaign data at equivalent latitudes are highlighted, particularly for CCN0.5.

P20-21: I would suggest moving L527-543 to the conclusions.

---

## Author Comment (AC1)

**Author response to referee comments**

Measurement Report: Understanding the seasonal cycle of Southern Ocean aerosols

Ruhi S. Humphries et al.,

**Response to Anonymous Referee #1**

We would like to thank referee #1 for the time taken and the thoughtful suggestions on our manuscript. We detail the improvements made to the paper in response to the reviewer's suggestions below:

- Minor suggestions – we agree that the inclusion of particle size distribution data at Macquarie Island is an important observation in this region and the authors of this study are working towards the establishment of these, together with other complementary measurements, at this remote location. Unfortunately, the scope of the described campaign only extended to the inclusion of $CN_{10}$ and CCN.
- L25 – we have updated this sentence to be along the lines of what the referee has suggested.
- L26-27 – this suggestion changes the meaning of the sentence. We have kept it as is.
- L27-28 – we have updated this sentence to better acknowledge that this region is not completely pollution free, but is the closest we have.
- L55 – we thank the authors for alerting us to this important study. We have included the published version of the paper in the citation list.
- L300 – while we agree that back trajectories are a useful tool in many cases, for this particular question, observations of radon concentrations are a better tool because they are a direct tracer of continental influence and don't suffer from the same high uncertainties that are present in back trajectory calculations in this region (which are driven by a paucity of observations feeding into the reanalysis datasets that drive the trajectory calculations).
- L319-320 – we have added several references to justify this statement.

**Response to Anonymous Referee #2**

We would like to thank reviewer 2 for their considered comments. They have been an asset to the study and have led to an improved manuscript. We have responded below to each:

Restructuring comments:

While we understand the reviewer's comments about restructuring the results section separate it by measured parameter, we feel that the structure present, where we have separated it by seasonal and latitudinal discussions, is equally valid. Referee #1 confirmed that the manuscript in the current form "follows a logical structure", and this has also been reviewed and agreed upon by all co-authors prior to submission. With regards to the seemingly contradictory conclusions, the examples that are provided by the referee have been addressed in response to specific comments below, so don't

present a potential contraction anymore. Consequent to these points, we have left the overall structure as is.

Specific comments:

- P4 Methods – we agree that a summary of the data availability would be a useful addition to the manuscript, but we extend this to beyond the referees suggestion of just Macquarie Island, and have added a table summarising the data availability from all stations and campaigns. An additional two sentences have been added at the conclusion of the first paragraph of the methods section to complement and introduce the new table.
- P6, L150-151 – this has been clarified in the text.
- P7-8, L195-198 – this is an excellent suggestion to help the reader understand the data availability and complements the addition of the table added to the methods in response to a previous comment. We have added a sentence at the end of section 2.2 that describes the baseline frequency.
- P8, L200-201 – this sentence has been reworded to improve clarity.
- P10, L271 – nice catch! We have updated CAPRICORN2 to CAPRICORN1 as per the referee's astute observation.
- P11 – Results and Discussion, general comments
  - Tempering the interpretation of certain observations by taking into account variability.
    - L344 – replaced "substantially" with "noticeably"
    - L467 – removed "significantly"
    - L504 – removed "substantially"
    - L516 – this instance is justified given the high sample number (both stations are in the thousands of hours of available data) and the obvious difference between the two stations. The text has been left as is.
    - L519 – this instance is justified given the difference between bins is over double, with tight distributions for all voyage data. In addition, the following sentence acknowledges the limited number of observations. The text has been left as is.
  - Discussion of various points:
    1. The difference of the length of the datasets is discussed briefly in the methods section. The choice of approximately 10 years of data was done "to enable a climatological comparison with the MQI data".
    2. The time lags between some datasets aren't of major impact on the results here because we are looking at data in a climatological sense, and campaign data are presented as individual data points on any figures. The years of long-term data utilised from SYO and KCG were chosen so as to overlap as much as possible with campaign data so as to minimize any long-term changes.
- P11, L309 – Corrected the typo by changing the second "is" to "as".
- P11, L310 – We have modified this sentence to clarify that KCG doesn't observe the highest concentrations all year around, but instead "for most of the year".
- P11, L315-316 – changed "September" to "August"
- P11, L315-316 – we have changed "although" to "while" to improve the clarity of the sentence.
- P13, L348-349 – we have reworded this part of the sentence to improve clarity.

- P11-14 – a further analysis of data from each year, in comparison with the 10 year climatology shown in the paper, shows that these features are consistent across almost all individual years. This is shown in the figure below. This figure hasn't been reproduced in the manuscript, however, we have added a sentence in the first paragraph of section 3.1 describing these results.

[Figure]

*Figure 1: Seasonal cycles of CN10 calculated from all available data at each station, plotted together with individual years. Monthly medians are shown in all cases, with the shading of the climatology representing the interquartile range.*

- P15, L394-395 – we agree that the majority of the peak is definitely in June, however we included May because the 75$^{th}$ percentile of the May data shows an increase in several parameters. We have altered the first sentence of this paragraph to reflect this, such that it now reads "… the significant spike in CCN0.5 concentrations at MQI, beginning in May, and peaking in June."
- P15, L424-426 – we have modified the sentence to include the importance of considering supersaturation in the discussion, and have explicitly outlined that we have utilised CCN measured at 0.5% supersaturation in the calculation of the ratio.
- P16, L445 – we have weakened this language, making it more quantitative, changing it from "many" to "around half".
- P16, L446 – while we agree that trajectories could add strength to this hypothesis, the authors feel that this is not a major result of this paper, and as such, is out of the scope of the current study. However, this is in line with previous studies that have utilised trajectory studies to analyse similar data. We have edited the sentence to add these citations to support this hypothesis.
- P17, L461-463 – we have reworded the sentence to improve wording.
- P19, L512-513 – We have added a few sentences in the previous paragraph to describe the difference between voyage and KCG data during the summer period, and also expanded the explanation of the autumn difference in the following paragraph.
- P20, L527, 528 – the improvements to previous discussions made in response to earlier comments by the referee improve the accuracy of this statement. However, we have edited this sentence acknowledge the difference.
- P20-21 – while we can see how these two sentences could be included in the conclusions, we feel that to discuss the points in the detail required, they are more suited to the discussion section where we can discuss in a more expansive form.